# VICON: Vision In-Context Operator Networks for Multi-Physics Fluid Dynamics Prediction

**Yadi Cao**[*][†]                                                                                  *yadicao95@ucla.edu*
*University of California, Los Angeles*
*University of California, San Diego*

**Yuxuan Liu**[*]                                                                                  *yxliu@ucla.edu*
*University of California, Los Angeles*

**Liu Yang**[†]                                                                                  *yangliu@nus.edu.sg*
*University of California, Los Angeles*
*National University of Singapore*

**Rose Yu**                                                                                  *roseyu@ucsd.edu*
*University of California, San Diego*

**Hayden Schaeffer**                                                                                  *hschaeffer@ucla.edu*
*University of California, Los Angeles*

**Stanley Osher**                                                                                  *sjo@ucla.edu*
*University of California, Los Angeles*

**Reviewed on OpenReview:** *https://openreview.net/forum?id=6V3YmHULQ3*

## Abstract

In-Context Operator Networks (ICONs) have demonstrated the ability to learn operators across diverse partial differential equations using few-shot, in-context learning. However, existing ICONs process each spatial point as an individual token, severely limiting computational efficiency when handling dense data in higher spatial dimensions. We propose *Vision In-Context Operator Networks* (VICON), which integrate vision transformer architectures to efficiently process 2D data through patch-wise operations while preserving ICON's adaptability to multi-physics systems and varying timesteps. Evaluated across three fluid dynamics benchmarks, VICON significantly outperforms state-of-the-art baselines DPOT and MPP, reducing the average last-step rollout error by 37.9% compared to DPOT and 44.7% compared to MPP, while requiring only 72.5% and 34.8% of their respective inference times. VICON naturally supports flexible rollout strategies with varying timestep strides, enabling immediate deployment in *imperfect measurement systems* where sampling frequencies may differ or frames might be dropped—common challenges in real-world settings—without requiring retraining or interpolation. In these realistic scenarios, VICON exhibits remarkable robustness, experiencing only 24.41% relative performance degradation compared to 71.37%-74.49% degradation in baseline methods, demonstrating its versatility for deployment in realistic applications. Our scripts for processing datasets and code are publicly available at `https://github.com/Eydcao/VICON`.

---

[*]Equal contribution.
[†]Work partially done while at UCLA.

# 1 Introduction

Machine learning has emerged as a powerful tool for solving Partial Differential Equations (PDEs). Traditional approaches primarily operate on discrete representations, with Convolutional Neural Networks (CNNs) excelling at processing regular grids Gupta & Brandstetter (2022), Graph Neural Networks (GNNs) handling unstructured meshes Sanchez-Gonzalez et al. (2020); Lino et al. (2022); Cao et al. (2023), and transformers capturing dependencies between sampling points within the same domains Cao (2021); Dang et al. (2022); Jiang et al. (2023). However, these discrete approaches face a fundamental limitation: they lack PDE context information (such as governing parameters) and consequently cannot generalize to new parameters or PDEs beyond their training set.

This limitation led to the development of "operator learning," where neural networks learn mappings from input functions to output functions, such as from initial/boundary conditions to PDE solutions. This capability was first demonstrated using shallow neural networks Chen & Chen (1995b;a), followed by specialized architectures including the widely adopted Deep Operator Network (DeepONet) Lu et al. (2021), Fourier Neural Operator (FNO) Li et al. (2021); Kovachki et al. (2023), and recent transformer-based operator learners Li et al. (2022; 2023); Ovadia et al. (2024). While these methods excel at learning individual parametric PDEs, they cannot generalize across different PDE types, necessitating costly retraining for new equations or even different timestep sizes.

Inspired by the success of large language models in multi-domain generation, researchers have explored "multi-physics PDE models" to address this single-operator limitation. Common approaches employ pre-training and fine-tuning strategies Chen et al. (2024); Herde et al. (2024); Zhang et al. (2024c), though these require substantial additional data (typically hundreds of frames) and fine-tuning before deployment. This requirement severely limits their practical applications in scenarios like online control and data assimilation, where rapid adaptation with limited data sampling is essential.

To eliminate the need for additional data collection and fine-tuning, several works have attempted to achieve few-shot or zero-shot generalization across multi-physics systems by incorporating explicit PDE information. For example, the PROSE approach Liu et al. (2024c;b); Sun et al. (2024a); Jollie et al. (2024) processes the symbolic form of governing equations using an additional transformer branch. Similarly, PDEformer Ye et al. (2024) converts PDEs into directed graphs as input for graph transformers, while others embed symbol-tokens directly into the model Lorsung et al. (2024). Despite these advances, such methods require explicit knowledge of the underlying PDEs, which is often unavailable when deploying models in new environments.

To further lower barriers for applying multi-physics models in real-world applications, the In-Context Operator Network (ICON) Yang et al. (2023a) offers a fundamentally different approach: ICON implicitly encodes system dynamics through a few input-output function pairs, then extracts dynamics from these pairs in an in-context fashion. This approach enables few-shot generalization without retraining or explicit PDE knowledge, while the minimal required input-output pairs can be readily collected during deployment. A single ICON model has demonstrated success in handling both forward and inverse problems across various ODEs, PDEs, and mean-field control scenarios.

However, ICON's computational efficiency becomes a critical bottleneck when handling dense data in higher dimensions. Specifically, ICON treats each sampled spatial point as an individual token, leading to quadratic computational complexity with respect to the number of points. This makes it computationally prohibitive for practical 2D and 3D applications—to date, only one instance of a 2D case using sparse data points has been demonstrated Yang et al. (2023a).

To address this limitation, we propose *Vision In-Context Operator Networks* (VICON), which leverage vision transformers to process 2D functions using an efficient patch-wise approach. Going beyond classic vision transformers that typically process patches from a single image, VICON extends the architecture to handle sequences of input-output function pairs, enabling "next function prediction" capabilities while preserving the benefits of in-context operator learning. The current implementation focuses on passive, autonomous PDEs; extension to non-autonomous systems with external forcing is left for future work.

Our contributions include:

- Development of VICON, a vision transformer-based in-context operator network for two-dimensional time-dependent PDEs that maintains the flexibility of in-context operator learning while efficiently handling dense data in higher dimensions.

- Comprehensive evaluation across diverse fluid dynamics systems comprising 879K frames with varying timestep sizes, demonstrating substantial improvements over state-of-the-art models MPP McCabe et al. (2023) and DPOT Hao et al. (2024) in both accuracy and efficiency. Our approach reduces the averaged last-step rollout error by 37.9% compared to DPOT and 44.7% compared to MPP, while requiring only 72.5% and 34.8% of their respective inference times.

- Enhanced flexibility supporting varying timestep strides and non-sequential measurements, offering substantial robustness in realistic scenarios with imperfect data collection, e.g. with missing frames. Under these scenarios, VICON exhibits only 24.41% relative performance degradation versus 71.37%-74.49% degradation in baseline methods.

## 2 Related Work

**Operator Learning.** Operator learning Chen & Chen (1995b;a); Li et al. (2021); Lu et al. (2021) addresses the challenge of approximating operators $G : U \to V$, where $U$ and $V$ are function spaces representing physical systems and differential equations, e.g., $G$ maps initial/boundary conditions or system parameters to corresponding solutions. Among the various operator learning approaches, DeepONets Lu et al. (2021; 2022) employ branch and trunk networks to independently process inputs and query points, while FNOs Li et al. (2021) leverage fast Fourier transforms to efficiently compute kernel integrations for PDE solutions in regular domains. These methods have been extended to incorporate equation information Wang et al. (2021); Li et al. (2024b), multiscale features Zhang et al. (2024a); Wen et al. (2022); Liu et al. (2024a), and adaptation to heterogeneous and irregular meshes Zhang et al. (2023; 2024b); Li et al. (2024a); Wu et al. (2024).

However, these approaches typically learn a single operator, requiring retraining when encountering different PDE types or timestep sizes. Our work aims to develop a unified model for multiple PDEs with few-shot generalization capabilities based on limited observation frames.

**Multi-Physics PDE Models.** Foundation models in natural language processing Brown et al. (2020); Touvron et al. (2023) and computer vision Ramesh et al. (2021) have demonstrated remarkable versatility across diverse tasks. Drawing inspiration from this paradigm, recent research has explored developing unified models for multiple PDEs in domains such as PDE discovery Schaeffer (2017) and computational fluid dynamics Wang et al. (2024a). Several approaches follow pre-training and fine-tuning strategies Chen et al. (2024); Herde et al. (2024), though these require additional resources for data acquisition and fine-tuning before deployment. In parallel, researchers have developed zero- or few-shot multi-physics models by incorporating additional information; for instance, the PROSE approach Liu et al. (2024c;b); Sun et al. (2024a); Jollie et al. (2024); Sun et al. (2024b) directly encodes symbolic information of PDEs into the model. Other methods achieving zero- or few-shot generalization include using physics-informed tokens Lorsung et al. (2024), representing PDE structures as graphs Ye et al. (2024), or conditioning transformers on PDE descriptions Zhou et al. (2025). However, all these techniques require prior knowledge of the PDEs as additional inputs. We select two state-of-the-art models, MPP McCabe et al. (2023) and DPOT Hao et al. (2024), as our baselines.

**In-Context Operator Networks.** ICONs Yang et al. (2023a;b); Yang & Osher (2024) learn operators by observing input-output function pairs, enabling few-shot generalization across various PDEs without requiring explicit PDE representations or fine-tuning. These networks have demonstrated success in handling forward and inverse problems for ODEs, PDEs, and mean-field control scenarios. However, ICONs face significant computational challenges when processing dense data, as they represent functions through scattered point tokens, resulting in quadratic computational complexity with respect to the number of sample points. This limitation has largely restricted their application to 1D problems, with only sparse sampling feasible in 2D cases Yang et al. (2023a). Our work addresses these limitations by introducing a vision transformer architecture that efficiently handles dense 2D data while preserving the benefits of in-context operator learning.

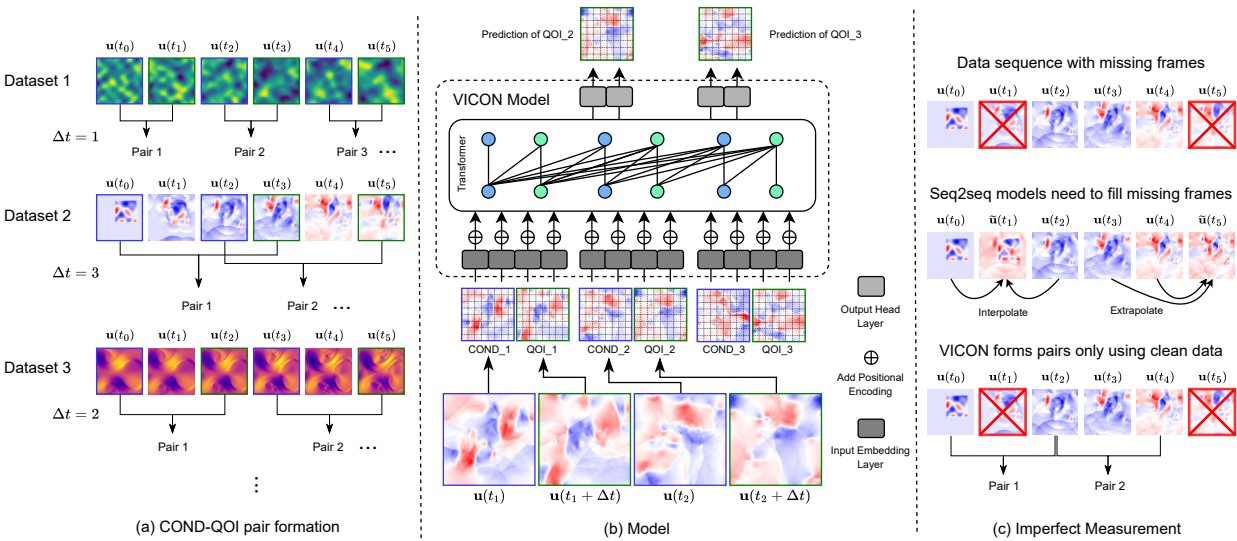

Figure 1: **VICON model overview.** (a) The formation process for conditions (COND) and quantities of interest (QOI) pairs. $\Delta t$ is randomly sampled during training. (b) Model illustration. The inputs to the model are pairs of COND and QOI, which are patchified and flattened before feeding into the transformer layers. The outputs, which represent different patches in the output frame, are transformed back to obtain the final predictions. (c) With imperfect temporal measurements, VICON forms pairs using only clean data, and does not need to fill missing frames.

## 3 Preliminaries

ICONs were introduced and developed in Yang et al. (2023a;b); Yang & Osher (2024). They adapt the in-context learning framework from large language models, aiming to train a model that can learn operators from prompted function examples.

Denote an ICON model as $\mathcal{T}_\theta$, where $\mathcal{T}$ is a transformer with trainable parameters $\theta$. The model takes input as a sequence comprising $I$ pairs of conditions ($\boldsymbol{c}$) and quantities of interest ($\boldsymbol{q}$) as $\{\langle \boldsymbol{c}_i, \boldsymbol{q}_i \rangle\}_{i=1}^{I}$, where each $\boldsymbol{c}, \boldsymbol{q}$ contains multiple tokens representing a single function. The model outputs tokens representing future functions at the positions of $\boldsymbol{c}$, i.e., "next function prediction" similar to "next token predictions" in LLMs. More precisely, for $J \in \{1, \ldots, I-1\}$, the model predicts $\tilde{\boldsymbol{q}}_{J+1}$ given the leading $J$ pairs and the condition $\boldsymbol{c}_{J+1}$:

$$\tilde{\boldsymbol{q}}_{J+1} = \mathcal{T}_\theta[\boldsymbol{c}_{J+1}; \{\langle \boldsymbol{c}_i, \boldsymbol{q}_i \rangle\}_{i=1}^{J}]. \tag{1}$$

For training parallelization, ICON uses a special causal attention mask to perform autoregressive learning (i.e., the model only sees the leading $J$ pairs and $\boldsymbol{c}_{J+1}$ when predicting $\tilde{\boldsymbol{q}}_{J+1}$) and enables output of all predictions within a single forward pass Yang et al. (2023b):

$$\{\tilde{\boldsymbol{q}}_i\}_{i=1}^{I} = \mathcal{T}_\theta[\{\langle \boldsymbol{c}_i, \boldsymbol{q}_i \rangle\}_{i=1}^{I}]. \tag{2}$$

Training loss is computed as the mean squared error (MSE) between the predicted states $\tilde{\boldsymbol{q}}_i$ and the ground truth states $\boldsymbol{q}_i$. To ensure that the model receives sufficient contextual information about the underlying dynamics, we only compute errors for the indices $i > I_{\min}$, effectively requiring at least $I_{\min}$ examples in context. This approach has shown empirical improvements in model performance.

After training, ICON can process a new set of $\langle \boldsymbol{c}, \boldsymbol{q} \rangle$ pairs in the forward pass to learn operators in-context and employ them to predict future functions using new conditions:

$$\tilde{\boldsymbol{q}}_k = \mathcal{T}_\theta[\boldsymbol{c}_k; \{\langle \boldsymbol{c}_i, \boldsymbol{q}_i \rangle\}_{i=1}^{J}], \tag{3}$$

where $J \in \{I_{\min}, \ldots, I-1\}$ is the number of in-context examples, $I_{\min}$ is the number of context examples exempted from loss calculation, and $k > I$ is the index of the condition for future prediction.

Importantly, all pairs in the same sequence must be formed with the same operator mapping (i.e., the same PDE and consistent timestep size within one sequence), while operators can vary across different sequences/rows in the training batch to enable the model's generalization ability to different combinations of PDEs and timestep sizes.

## 4  Methodology

### 4.1  Problem Setup

We consider the forward problem for multiple time-dependent PDEs that are *temporally homogeneous Markovian*, defined on domain $\Omega \subseteq \mathbb{R}^2$ with solutions represented by $\mathbf{u}(\boldsymbol{x}, t) : \Omega \times [0, T] \to \mathbb{R}^c$, where $c$ is the number of channels. Given the initial $I_0$ frames, $\{\mathbf{u}_i = \mathbf{u}(\cdot, t_i) \mid i = 1, \ldots, I_0\}$, the task is to predict future solutions.

Each solution frame is discretized as a three-dimensional tensor $\mathbf{u}_t = \mathbf{u}(\cdot, t) \in \mathbb{R}^{N_x \times N_y \times c}$, where $N_x$ and $N_y$ are the spatial grid sizes. Due to the homogeneous Markovian property, given the initial data $\{\mathbf{u}_t \mid t \geq 0\}$ from some PDE indexed by $i_p$, for a fixed $\Delta t$, there exists an operator $\mathcal{L} = \mathcal{L}_{\Delta t}^{(i_p)}$ that maps frame $\mathbf{u}_t$ to frame $\mathbf{u}_{t+\Delta t}$ for any $t \geq 0$. Our goal is to train a model that learns these operators $\mathcal{L}_{\Delta t}^{(i_p)}$ from the sequence of function pairs generated from the initial frames. Notably, even within the same dataset and fixed $\Delta t$, different trajectories can exhibit different dynamics (e.g., due to different Reynolds numbers $Re$). These variations are implicitly captured by the function pairs in our framework.

### 4.2  Vision In-Context Operator Networks

The original ICONs represent functions as scattered sample data points, where each data point is projected as a single token. For higher spatial dimensions, this approach necessitates an excessively large number of sample points and extremely long sequences in the transformer. Due to the inherent quadratic complexity of the transformer, this approach becomes computationally infeasible for high-dimensional problems.

Inspired by the Vision Transformer (ViT) Dosovitskiy et al. (2021), we address these limitations by dividing the physical fields into patches, where each patch is flattened and projected as a token. This approach, which we call Vision In-Context Operator Networks (VICON), addresses the computational limitation of treating each spatial point as an individual token, which leads to quadratic computational complexity with respect to the number of points. This patch-wise approach has its forward process illustrated in Figure 1(b) and detailed in the following.

First, the input $\boldsymbol{u}_t$ and output $\boldsymbol{u}_{t+\Delta t}$ functions are divided into patches $\{\boldsymbol{C}^k \in \mathbb{R}^{R_x \times R_y \times c}\}_{k=1}^{N_c}$ and $\{\boldsymbol{Q}^l \in \mathbb{R}^{R_x \times R_y \times c}\}_{l=1}^{N_q}$, where $R_x, R_y$ are the resolution dimensions of the patch and $N_c$ and $N_q$ are the number of patches for the input and output functions, respectively. While $N_q$ can differ from $N_c$ (for instance, when input functions require additional boundary padding, resulting in $N_c > N_q$), we maintain $N_c = N_q$ throughout our experiments. For notational clarity, we add the subscript $i \in \{1, \ldots, I\}$ to denote the pair index, where $\boldsymbol{C}_i^k$ represents the $k$-th patch of the input function and $\boldsymbol{Q}_i^l$ represents the $l$-th patch of the output function in the $i$-th pair. These patches are then projected into a unified $d$-dimensional latent embedding space using a shared learnable linear function $f_\phi : R_x \times R_y \times c \to \mathbb{R}^d$:

$$\hat{\boldsymbol{c}}_i^k = f_\phi(\boldsymbol{C}_i^k), \quad \hat{\boldsymbol{q}}_i^l = f_\phi(\boldsymbol{Q}_i^l), \tag{4}$$

where $k = 1, \ldots, N_c$ and $l = 1, \ldots, N_q$ are the index of patches. We use the same projection $f_\phi$ for both conditions and QoIs because in our setup, both represent the same physical fields at different timesteps; sharing the projection enforces consistency in latent space regardless of whether a frame serves as context or target.

We inject two types of learnable positional encoding before feeding the embeddings into the transformer: (1) patch positional encodings to indicate relative patch positions inside the whole domain, denoted as $\mathbf{E}_p \in \mathbb{R}^{N_p \times d}$, where $N_p = \max\{N_c, N_q\}$ and the encoding varies per patch (broadcast across points within each patch); (2) function positional encodings to indicate whether a patch belongs to an input function

(using $\mathbf{E}_c \in \mathbb{R}^{I \times d}$) or output function (using $\mathbf{E}_q \in \mathbb{R}^{I \times d}$), as well as their indices in the sequence, where the encoding varies per condition/QoI in each pair (broadcast across patches and points of that condition/QoI):

$$\boldsymbol{c}_i^k = \hat{\boldsymbol{c}}_i^k + \mathbf{E}_p(k) + \mathbf{E}_c(i), \quad \boldsymbol{q}_i^l = \hat{\boldsymbol{q}}_i^l + \mathbf{E}_p(l) + \mathbf{E}_q(i). \tag{5}$$

The embeddings $\boldsymbol{c}_i^k$ and $\boldsymbol{q}_i^l$ are then concatenated to form the input sequence for the transformer:

$$\mathbf{c}_1^1 \ldots \mathbf{c}_1^{N_c}, \mathbf{q}_1^1 \ldots \mathbf{q}_1^{N_q}, \ldots, \mathbf{c}_I^1 \ldots \mathbf{c}_I^{N_c}, \mathbf{q}_I^1 \ldots \mathbf{q}_I^{N_q}.$$

To support autoregressive prediction similar to Equation (2), VICON employs an alternating-sized (in the case where $N_c \neq N_q$) block causal attention mask, as opposed to the conventional triangular causal attention mask as in mainstream generative large language models Brown et al. (2020) and large vision models Bai et al. (2024). The mask is defined as follows:

$$\mathbf{M} = \begin{bmatrix} \mathbf{1}_{N_c, N_c} & \mathbf{0} & \cdots & \mathbf{0} & \mathbf{0} \\ \mathbf{1}_{N_q, N_c} & \mathbf{1}_{N_q, N_q} & \cdots & \mathbf{0} & \mathbf{0} \\ \vdots & \vdots & \ddots & \vdots & \vdots \\ \mathbf{1}_{N_c, N_c} & \mathbf{1}_{N_c, N_q} & \cdots & \mathbf{1}_{N_c, N_c} & \mathbf{0} \\ \mathbf{1}_{N_q, N_c} & \mathbf{1}_{N_q, N_q} & \cdots & \mathbf{1}_{N_q, N_c} & \mathbf{1}_{N_q, N_q} \end{bmatrix} \tag{6}$$

where $\mathbf{1}_{m \times n}$ denotes an all-ones matrix of dimension $m \times n$, and $\mathbf{0}$ represents a zero matrix of the corresponding dimensions.

After obtaining the output tokens from Equation (2), we extract the tokens corresponding to the input patch indices $\mathbf{c}_i^1, \ldots, \mathbf{c}_i^{N_c}$ and denote them as $\tilde{\boldsymbol{q}}_i^1, \ldots, \tilde{\boldsymbol{q}}_i^{N_c}$. These tokens are then projected back to the original physical space using a shared learnable linear function $g_\psi : \mathbb{R}^d \to \mathbb{R}^{R_x \times R_y \times c}$:

$$\tilde{\boldsymbol{Q}}_i^l = g_\psi(\tilde{\boldsymbol{q}}_i^l), \tag{7}$$

which predicts $\boldsymbol{Q}_i^l$.

### 4.3 Prompt Normalization

To address the varying scales across different channels and prompts (i.e., sequences of pairs $\{\langle \boldsymbol{c}_i, \boldsymbol{q}_i \rangle\}_{i=1}^I$), we normalize the data before feeding them into the model. A crucial requirement is to consistently normalize functions within the same sequence, to ensure that the same operator is learned. Denoting the normalization operators for $\boldsymbol{c}$ and $\boldsymbol{q}$ as $\mathcal{N}_c$ and $\mathcal{N}_q$ respectively, and the operator in the original space as $\mathcal{L}$, the operator in the normalized space $\mathcal{L}'$ follows:

$$\mathcal{L}'(\mathcal{N}_c(\mathbf{u})) = \mathcal{N}_q(\mathcal{L}(\mathbf{u})). \tag{8}$$

In this work, we simply set $\mathcal{N}_c = \mathcal{N}_q$, mainly because our maximum timestep stride $s_{\max} = 5$ is relatively small (i.e., the scale distribution does not change dramatically). Specifically, given input sequence $\{\langle \boldsymbol{c}_i, \boldsymbol{q}_i \rangle\}_{i=1}^I$, we compute the channel-wise mean $\mu = \mathtt{mean}(\boldsymbol{c}_1, \boldsymbol{c}_2, \ldots, \boldsymbol{c}_I) \in \mathbb{R}^d$ and standard deviation $\sigma = \mathtt{std}(\boldsymbol{c}_1, \boldsymbol{c}_2, \ldots, \boldsymbol{c}_I) \in \mathbb{R}^d$ of all conditions $\{\boldsymbol{c}_i\}_{i=1}^I$, which are then used to normalize both the conditions and QoIs. Training MSE loss is computed in the normalized space. To avoid division by zero, we set a minimum threshold of $10^{-4}$ for the standard deviation.

### 4.4 Datasets and Data Augmentation

We evaluate on three fluid dynamics datasets representing different physical regimes: 1) PDEArena-Incomp Gupta & Brandstetter (2022) (incompressible Navier-Stokes equations), containing 2,496/608/608 trajectories (train/valid/test) with 56 timesteps each; 2) PDEBench-Comp-HighVis Takamoto et al. (2022) (compressible Navier-Stokes with high viscosity), containing 40,000 trajectories of 21 timesteps each; and 3) PDEBench-Comp-LowVis Takamoto et al. (2022) (compressible Navier-Stokes with numerically

zero viscosity), containing 4,000 trajectories of 21 timesteps each. For PDEArena-Incomp, we follow the original data split in Gupta & Brandstetter (2022). For PDEBench-Comp-HighVis and PDEBench-Comp-LowVis, we randomly split trajectories in $80\%/10\%/10\%$ proportions for training/validation/testing, respectively. For all datasets during rollout, we set $I_0 = 10$, i.e., we predict trajectories up to the last timestep given the initial 10 frames.

The temporally homogeneous Markovian property (Section 4.1) enables natural data augmentation for training and flexible rollout strategy for inference (Section 4.5): by striding with larger timesteps to reduce the number of autoregressive steps in long-term prediction tasks: $\Delta t = s\Delta\tau_{i_p} \mid s = 1, 2, \ldots, s_{\max}$, where $\tau_{i_p}$ is the timestep size for recording in trajectory $i_p$. During training, for each trajectory in the training set, we first sample a stride size $s \sim \mathcal{U}\{1, \ldots, s_{\max}\}$, then form $\langle \mathbf{u}_t, \mathbf{u}_{t+s\Delta\tau} \rangle$ pairs for the corresponding operator. We illustrate this augmentation process in Figure 1(a), where different strides are randomly sampled during dataloading.

Additional dataset details appear in Appendix A.

## 4.5 Inference with Flexible Strategies

As mentioned in Section 4.4, VICON can make predictions with varying timestep strides. Given $I_0$ initial frames $\{\mathbf{u}_i\}_{i=1}^{I_0}$, we can form in-context example pairs $\{\langle \mathbf{u}_i, \mathbf{u}_{i+s} \rangle\}_{i=1}^{I_0-s}$. For $j \geq 1$, we set the question condition ($\boldsymbol{c}_k$ in Equation (3)) as $\boldsymbol{u}_j$ to predict $\boldsymbol{u}_{j+s}$, enabling $s$-step prediction.

For long-term rollout, we employ VICON in an autoregressive fashion. The ability to predict with different timestep strides enables various rollout strategies. We explore two natural approaches: single-step rollout and flexible-step rollout, where the latter advances in larger strides to reduce the number of rollout steps.

For single-step rollout, we simply follow the autoregressive procedure with $s = 1$. Flexible-step rollout involves a more sophisticated approach: given a maximum prediction stride $s_{\max}$, we first make sequential predictions with a gradually growing $s = 1$ to $s_{\max}$ using $\mathbf{u}_{I_0}$ as the question condition, obtaining $\{\mathbf{u}_{I_0+s}\}_{s=1}^{s_{\max}}$. Using each frame in this sequence as a question condition, we then make consistent $s_{\max}$-step predictions while preserving all intermediate frames. This flexible-step strategy follows the approach in Yang & Osher (2024).

This strategy is also applicable to imperfect measurement cases where partial input frames are missing, e.g., due to sensor device error. In this case, we can still form pairs from the remaining frames, with minor adjustments to the strategy generation algorithms, as shown in Figure 1(c).

More details (including algorithms and examples) on rollout strategies are provided in Appendix C.1 (for perfect measurements) and in Appendix C.2 (for imperfect measurements).

## 4.6 Evaluation Metric

We evaluate the rollout accuracy of VICON using two different strategies described in Section 4.5. The evaluation uses relative and absolute $L^2$ errors between the predicted and ground truth frames, starting from the frame $I_0 + 1$. For relative scaling coefficients, we use channel-wise scaling standard deviation $\sigma$ of ground truth frames, which vary between different prompts.

## 5 Experimental Results

We benchmark VICON against two state-of-the-art sequence-to-sequence models: DPOT Hao et al. (2024) (122M parameters) and MPP McCabe et al. (2023) (AViT-B architecture, 116M parameters), as well as Specialist U-Net baselines Gupta & Brandstetter (2022) trained individually per dataset. Implementation details for all baselines are provided in Appendix B. The vanilla ICON, which would require processing over 114K tokens per frame ($128\times128\times7$), exceeds our available GPU memory and is thus computationally infeasible for direct comparison on these datasets.

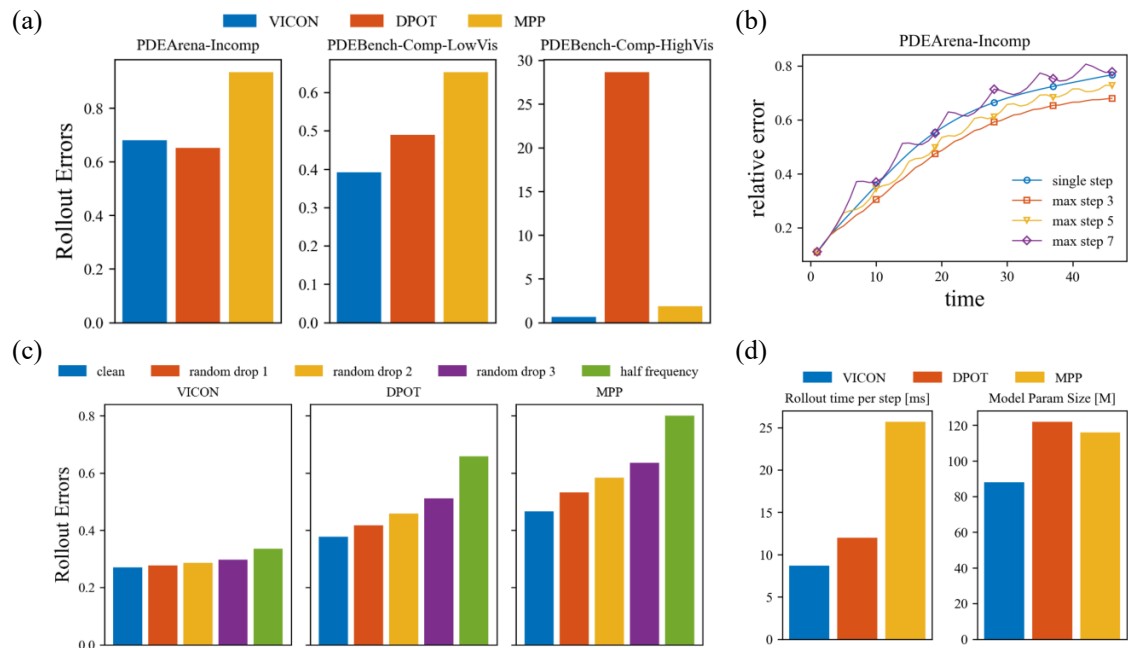

Figure 2: **Main experiment results.** (a) Last step rollout errors on 3 datasets. VICON outperforms MPP on all datasets and outperforms DPOT on 2 datasets. (b) VICON allows flexible rollout strategies to reduce error accumulation and demonstrates stride extrapolation. (c) VICON is robust to imperfect temporal measurements, while MPP and DPOT suffer from performance degradation. (d) VICON is smaller in size and has faster rollout time per step.

For evaluation, we use both absolute and relative $L^2$ RMSE (Section 4.6) on rollout predictions. Since VICON offers the flexibility of predictions with different timestep strides, we evaluate our single trained model using both single-step and flexible-step rollout strategies (Section 4.5) during inference.

For conciseness, we present summarized plots and tables in the main text, and defer complete results and visualizations to Appendix D. Ablation studies examining key design choices of VICON—including patch resolution, positional encoding, and context length, as well as an alternative CNN-based architecture—are presented in Appendix D.1. The checkerboard artifacts (e.g., in Figure 9) of the PDEBench-Comp-HighVis dataset are discussed in Appendix D.2.

Table 1: **Summary of Rollout Relative $L^2$ Error (scaled by std)** across different methods and datasets. The best results are highlighted in bold. For flexible step rollout, step 3 works best for the PDEArena-Incomp dataset. Specialist U-Net is trained individually per dataset with instance normalization matching VICON's preprocessing.

| Rollout Relative $L^2$ Error | Case | Ours (single step) | Ours (flexible step) | DPOT | MPP | Specialist U-Net |
|---|---|---|---|---|---|---|
| Last step [1e-2] | PDEArena-Incomp | 76.77 | 68.03 | 65.27 | 93.52 | **48.8** |
| | PDEBench-Comp-LowVis | **39.11** | **39.11** | 48.92 | 65.32 | 55.9 |
| | PDEBench-Comp-HighVis | 61.41 | 61.41 | 2866 | 185.3 | **39.9** |
| All average [1e-2] | PDEArena-Incomp | 56.27 | 48.50 | 41.20 | 55.95 | **27.9** |
| | PDEBench-Comp-LowVis | **27.08** | **27.08** | 37.72 | 46.68 | 35.7 |
| | PDEBench-Comp-HighVis | 30.06 | 30.06 | 821.9 | 72.37 | **19.9** |

### 5.1 Superior Performance on Long-term Rollout Predictions

As demonstrated in Figure 2(a), Table 1, and Figure 3, VICON consistently outperforms baseline methods on long-horizon predictions across all benchmarks—with the exception of DPOT on the PDEArena-Incomp dataset, where our performance is comparable. Overall, VICON achieves an average

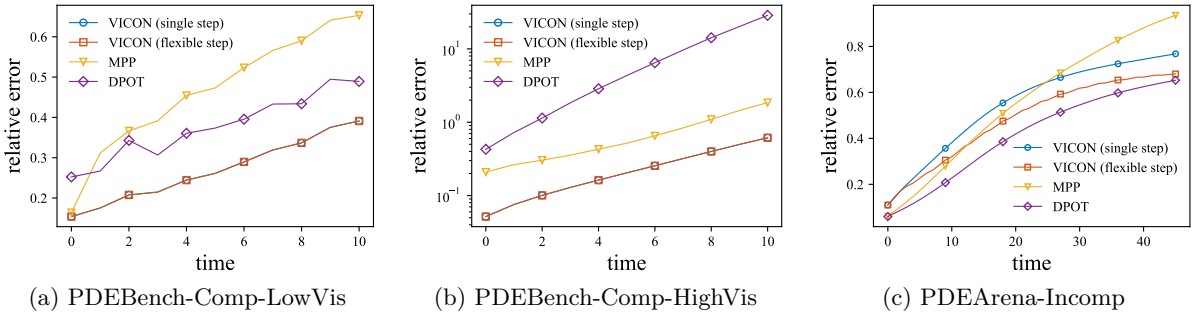

Figure 3: **Comparison of rollout errors (scaled by std) across different datasets and models.** We show errors for two VICON rollout strategies: single step rollout and flexible rollout strategy. For flexible step, step 3 works optimally for the PDEArena-Incomp dataset, while step 1 works best for PDEBench-Comp-LowVis and PDEBench-Comp-HighVis.

reduction in relative $L^2$ RMSE at the final timestep of 37.9% compared to DPOT and 44.7% compared to MPP.

Notably, DPOT exhibits exceptionally poor performance on PDEBench-Comp-HighVis, with an 821.9% error compared to our 30.06%. Our visualization of failure cases reveals that DPOT struggles with trajectories with small pressure values compared to the dataset average. This may stem from DPOT's lack of prompt normalization, as compressible flow's pressure channel exhibits large magnitude variations.

For the PDEArena-Incomp dataset, while VICON slightly underperforms DPOT and MPP initially, it quickly surpasses MPP and achieves comparable performance to DPOT for longer-step rollouts. This demonstrates VICON's robustness in long-term predictions. Despite DPOT's marginally better performance here, its poor performance on the PDEBench-Comp-HighVis dataset limits its applicability in multi-physics settings.

To further contextualize VICON's performance, we compare against Specialist U-Net baselines (U-Net-Mod64 from PDEArena Gupta & Brandstetter (2022)) trained individually per dataset. The only modification we made is adding instance normalization matching VICON's preprocessing, which we found essential for stabilizing training on PDEBench-Comp-LowVis and PDEBench-Comp-HighVis datasets. As shown in Table 1, while the specialist U-Net achieves strong performance on individual datasets it was optimized for, VICON remains competitive across all datasets and even outperforms the specialist on PDEBench-Comp-LowVis, demonstrating the advantage of our multi-physics approach in handling diverse physical systems without per-dataset architectural engineering or retraining.

More detailed results are provided in Table 13 and Table 14 in Appendix D.

## 5.2 Flexible Rollout Strategies

As demonstrated in Figure 2(b) and Figure 5, VICON can select appropriate timestep strides based on each dataset's characteristics. While PDEBench-Comp-LowVis and PDEBench-Comp-HighVis perform best with single-step rollout, PDEArena-Incomp achieves optimal results with a stride of 3 using our flexible-step rollout strategy.

This dataset-dependent performance reflects the characteristics of each dataset and the challenges of learning larger timestep strides. PDEBench-Comp-LowVis and PDEBench-Comp-HighVis record data with larger timestep sizes, making the learning of larger stride predictions more challenging due to increased dynamics complexity. Additionally, the fewer number of total rollout steps (even with stride=1) in these datasets cancels out the benefits of reducing rollouts when using multi-stride. Conversely, PDEArena-Incomp records data with smaller timesteps, making both single- and multi-stride prediction learning more feasible. Under this condition, multi-stride reduces the total rollout steps and error accumulation, achieving an 11.4% error reduction with a balanced stride ($s_{\max} = 3$).

When evaluated with unseen strides ($s_{\max} = 7$), VICON maintains comparable performance to single-step strategies for PDEArena-Incomp, indicating that it extracts underlying operators through context pairs rather than memorizing dynamics. This generalization capability provides tolerance when deployed to real-world settings where device sampling rates often differ from a fixed training set—an advantage we further demonstrate in the next section.

### 5.3 Importance of In-Context Learning

To validate the importance of in-context learning, we compare VICON against a non-in-context ViT baseline (with similar architecture: 162M parameters, 12 layers, hidden dimension 1024 for attention, and 4096 for FFN, time stride $\Delta t = 1$ for training) on the PDEArena-Incomp dataset. The results in Table 2 demonstrate that vanilla ViT cannot generalize to different timestep strides without in-context examples:

Table 2: Non-in-context ViT baseline comparison on PDEArena-Incomp dataset

| Model | Param Size | Stride 1 Error (1e-2) | Stride 2 Error (1e-2) |
|---|---|---|---|
| VICON (Last step) | 88 M | 76.77 | 68.03 |
| ViT (Last step) | 162 M | 73.94 | 95.03 |
| VICON (All average) | 88 M | 56.27 | 48.50 |
| ViT (All average) | 162 M | 50.17 | 63.15 |

The significant performance degradation (95.03 vs 73.94 for last step error) demonstrates that vanilla ViT cannot generalize to different $\Delta t$ without in-context examples, validating our approach's effectiveness.

Additionally, we evaluated the impact of incorrect context pairs on VICON's performance using the PDEArena-Incomp dataset (Table 3).

Table 3: Impact of incorrect context pairs on VICON performance

| Experiment Setting | Average Rollout Error |
|---|---|
| Correct (stride 1) | 0.55 |
| Mix context pairs with stride 1 and 2 | 0.88 |
| Context pairs with stride 2 | 0.92 |
| Random noise context pairs | 1.57 |

The clear performance degradation with incorrect context pairs confirms that VICON effectively utilizes in-context information rather than simply memorizing patterns.

### 5.4 Robustness to Imperfect Temporal Measurements

Real-world experimental measurements frequently suffer from imperfections—sampling rates may differ from the training set, and frames can be missing due to device errors. Sequence-to-sequence models like MPP and DPOT require either retraining with new data or interpolation that introduces noise. In contrast, VICON elegantly handles such imperfections by forming context pairs from available frames only (Figure 1(c)). The algorithm for generating pairs with imperfect measurements appears in Appendix C.2.

We evaluate VICON against baselines on all benchmarks under two scenarios: (1) half sampling rate (dropping every other frame) and (2) random frame dropping (removing 1-3 frames per sequence). Results for the PDEBench-Comp-LowVis dataset in Figure 2(c) (with complete results in Figure 8) show that VICON experiences only 24.41% relative performance degradation compared to 71.37%-74.49% in baselines—demonstrating remarkable robustness to measurement imperfections.

### 5.5 Turbulence Kinetic Energy Analysis

Turbulence kinetic energy (TKE), defined as $\frac{1}{2}(\overline{\widetilde{u}_x^2} + \overline{\widetilde{u}_y^2})$, is a critical metric for quantifying a model's ability to capture turbulent flow characteristics. Here, $\widetilde{\boldsymbol{u}} = \boldsymbol{u} - \overline{\boldsymbol{u}}$ denotes the fluctuation of velocity from its statistical equilibrium state $\overline{\boldsymbol{u}}$.

Within our datasets, only a subset of PDEBench-Comp-LowVis, specifically those initialized with fully developed turbulent fields (see Appendix D.5 of Takamoto et al. (2022)), is suitable for TKE analysis. After filtering these entries, we compare the mean absolute error (MAE) of the TKE between VICON, DPOT, and MPP. Our approach achieves a TKE error of 0.016, significantly outperforming MPP's error of 0.049, while achieving performance comparable to DPOT's error of 0.012. Visualizations of TKE errors are presented in Figure 6 in Appendix D.

### 5.6 Benefits of Multi-Physics Joint Training

We empirically examine whether VICON benefits from multi-physics training by comparing two strategies: (1) training a single model *jointly* on all three datasets versus (2) training *separate* models specialized for individual datasets. For fair comparison, we maintained identical batch sizes across all trainings while adjusting the steps for separate training to be slightly more than one-third of the joint training duration, ensuring **comparable total computational costs** between approaches.

As shown in Figure 7, joint training significantly outperforms separate training across all three datasets. We attribute this performance gain to the underlying physical similarities across these fluid dynamics problems, where exposure to diverse yet related flow patterns enhances the model's generalization ability.

### 5.7 Context Sampling Strategy

By default, VICON forms context pairs from initial frames within a single trajectory, which we term *intra-trajectory* context. This design choice is motivated by practical online deployment scenarios where only a few initial measurements are available, and system parameters (e.g., viscosity, forcing terms) may be unknown—making retrieval of historical trajectories with the same underlying operator infeasible.

As a theoretical question, we additionally investigate whether VICON can leverage *cross-trajectory* context, i.e., context pairs from different trajectories governed by the same operator. As detailed in Appendix D.3, VICON trained with cross-trajectory context achieves similar performance with mixed contexts during rollout (only 0.012 increase in average error). This demonstrates the model's capability to leverage cross-trajectory information when available. Nevertheless, we argue that intra-trajectory context remains more practical as it avoids the need for maintaining trajectory databases and online operator identification.

### 5.8 Computational Efficiency

In Figure 2(d) and Table 16, we compare the computational resources required by VICON and baselines. Our model demonstrates superior efficiency across multiple metrics. Compared to MPP, VICON requires approximately one-third of the inference time per frame while using 75% of the total parameters. VICON also outperforms DPOT, requiring 28% fewer parameters and 28% less inference time.

It is important to note that, in theory, VICON incurs approximately $4\times$ computational cost compared to non-paired, sequence-to-sequence baselines when all other settings (e.g., patch size) are identical, as it processes both condition and QoI, doubling the total token length with quadratic attention complexity. However, this computational overhead is justified by the significant performance improvements and unique capabilities (handling imperfect measurements, flexible timestep strides). Additionally, this cost can be reduced to $2\times$ (linear) at inference time using KV caching, where all context pairs except the final pair remain fixed across rollout steps, allowing their key-value multiplications to be cached.

It is important to note that while post-training (such as fine-tuning) on a sequence-to-sequence model may achieve the same accuracy on one physical condition and save this $2\times$ cost overhead, the fine-tuned model

loses the ability to predict accurately when physical parameters may change on-the-fly during deployment, while VICON provides superior flexibility.

## 6 Conclusion and Future Work

We present VICON, a vision-transformer-based in-context operator network that efficiently processes dense physical fields through patch-wise operations. VICON overcomes the computational burden of original in-context operator networks in higher spatial dimensions while preserving the flexibility to extract multi-physics dynamics from few-shot contexts. Our comprehensive experiments demonstrate that VICON achieves superior performance in long-term predictions, reducing the average last-step rollout error by 37.9% compared to DPOT and 44.7% compared to MPP, while requiring only 72.5% and 34.8% of their respective inference times. The model supports flexible rollout strategies with varying timestep strides, enabling natural application to imperfect real-world measurements where sampling frequencies differ or frames are randomly dropped—scenarios where VICON experiences only 24.41% relative performance degradation compared to 71.37%-74.49% degradation in baseline models.

Despite these advances, several challenges remain for future investigation. We empirically showed in Section 5.6 the benefits of multi-physics training, yet scaling to larger and more diverse datasets presents practical challenges. Specifically, generating high-fidelity physics simulation data across multiple domains requires substantial computational resources and specialized domain expertise including mesh generation, physical modeling, and numerical solver integration.

Furthermore, the current approach does not yet extend to 3D applications, as token sequence length would grow cubically, exceeding our computational budget. Potential remedies for 3D scaling include: (1) using VAE-like techniques/transformers to compress and evolve the fields in the latent space, then applying attention mechanisms to decode at any location Wang et al. (2024b); Li et al. (2023), and (2) employing crop-like methods to learn evolution of fixed-width windows similar to CNNs. These approaches could potentially be combined to address the cubic scaling challenge.

Since VICON is an autoregressive model, it still suffers from error accumulation of the final pair's condition frame during rollout. Future work could explore resolving error accumulation through techniques such as injecting Gaussian noise or using push-forward methods Brandstetter et al. (2021).

The channel-union approach is also limited when incorporating domains beyond fluid dynamics with fundamentally different state variables. Recent advances in inter-channel attention mechanisms Holzschuh et al. (2025) offer promising directions for handling channel scaling issues when expanding to more diverse physics systems.

Finally, adapting VICON to handle irregular domains such as graphs or meshes would broaden its applications to areas like solid mechanics and molecular dynamics. Techniques such as geometric deep learning extensions, masked patch modeling, or point-cloud modeling are potential future directions to handle irregular geometries. The original ICON framework has demonstrated success across various problem types including inverse problems and steady-state tasks Yang et al. (2023a), suggesting potential for extending VICON beyond forward temporal prediction. Addressing these challenges will advance the promising paradigm of in-context learning for a broader range of physical systems.

### Acknowledgments

Yadi Cao and Rose Yu are supported in part by the US Army Research Office under Army-ECASE award W911NF-23-1-0231, the US Department of Energy, Office of Science, IARPA HAYSTAC Program, CDC-RFA-FT-23-0069, DARPA AIE FoundSci, DARPA YFA, NSF grants #2205093, #2100237,#2146343, and #2134274. Yuxuan Liu and Hayden Schaeffer are supported in part by AFOSR MURI FA9550-21-1-0084 and NSF DMS 2427558. Liu Yang is supported by the NUS Presidential Young Professorship. Stanley Osher is partially funded by STROBE NSF STC887 DMR 1548924, AFOSR MURI FA9550-18-502 and ONR N00014-20-1-2787.

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

# A Dataset Details

## A.1 PDEArena-Incomp Dataset

The incompressible Navier-Stokes dataset comes from PDEArena Gupta & Brandstetter (2022). The data are generated from the equation

$$\partial_t \boldsymbol{u} + \boldsymbol{u} \cdot \nabla \boldsymbol{u} = -\nabla p + \mu \Delta \boldsymbol{u} + \boldsymbol{f}, \tag{9}$$
$$\nabla \cdot \boldsymbol{u} = 0. \tag{10}$$

The space-time domain is $[0, 32]^2 \times [18, 105]$ where $dt = 1.5$ and the space resolution is $128 \times 128$. A scalar particle field is being transported with the fluids. The velocity fields satisfy Dirichlet boundary conditions, and the scalar field satisfies Neumann boundary conditions. The forcing term $\boldsymbol{f}$ is randomly sampled. The quantities of interest are the velocities and the scalar particle field.

## A.2 PDEBench-Comp-HighVis and PDEBench-Comp-LowVis Datasets

The PDEBench-Comp-HighVis and PDEBench-Comp-LowVis datasets come from PDEBench Compressible Navier-Stokes dataset Takamoto et al. (2022). The data are generated from the equation

$$\partial_t \rho + \nabla \cdot (\rho \boldsymbol{u}) = 0, \tag{11}$$
$$\rho(\partial_t \boldsymbol{u} + \boldsymbol{u} \cdot \nabla \boldsymbol{u}) = -\nabla p + \eta \Delta \boldsymbol{u} + (\zeta + \eta/3)\nabla(\nabla \cdot \boldsymbol{u}), \tag{12}$$
$$\partial_t \left( \varepsilon + \frac{\rho u^2}{2} \right) = -\nabla \cdot \left( \left( \varepsilon + p + \frac{\rho u^2}{2} \right) \boldsymbol{u} - \boldsymbol{u} \cdot \sigma' \right). \tag{13}$$

The space-time domain is $\mathbb{T}^2 \times [0, 1]$ where $dt = 0.05$. The datasets contain different combinations of shear and bulk viscosities. We group the ones with larger viscosities into the PDEBench-Comp-HighVis dataset, and the ones with extremely small (1e-8) viscosities into the PDEBench-Comp-LowVis dataset. The PDEBench-Comp-HighVis dataset has space resolution $128 \times 128$. The PDEBench-Comp-LowVis dataset has raw space resolution $512 \times 512$ and is downsampled to $128 \times 128$ through average pooling for consistency. The quantities of interest are velocities, pressure, and density.

## A.3 QOI Union and Channel Mask

Since each dataset contains different sets of quantities of interest, we take their union to create a unified representation. The unified physical field has 7 channels in total, with the following ordering:

1. Density ($\rho$)

2. Velocity in the x direction ($u_x$)

3. Velocity in the y direction ($u_y$)

4. Pressure ($P$)

5. Vorticity ($\omega$)

6. Passively transported scalar field ($S$)

7. Node type indicator (0: interior node, 1: boundary node)

For each dataset, we use a channel mask to indicate its valid fields and only calculate loss on these channels. The node type channel is universally excluded from loss calculations across all datasets.

Table 4: Model Configuration Details

| | |
|---|---|
| *Patch Configuration* | |
| Input patch numbers | $8 \times 8$ |
| Output patch numbers | $8 \times 8$ |
| Patch resolution | $16 \times 16$ |
| *Positional Encodings Shapes* | |
| Patch positional encodings | $[64, 1024]$ |
| Function positional encodings | $[20, 1024]$ |
| *Transformer Configuration* | |
| Hidden dimension | 1024 |
| Number of attention heads | 8 |
| Feedforward dimension | 2048 |
| Number of layers | 10 |
| Dropout rate | 0.0 |
| Number of COND & QOI pairs | 10 |
| Number of QOI exempted from loss calculation | 5 |

Table 5: Optimization Hyperparameters

| Parameter | Value |
|---|---|
| *Learning Rate Schedule* | |
| Scheduler | Cosine Annealing with Linear Warmup |
| Peak learning rate | $1 \times 10^{-4}$ |
| Final learning rate | $1 \times 10^{-7}$ |
| Warmup steps | 20,000 |
| Total steps | 200,000 |
| *Optimization Settings* | |
| Optimizer | AdamW |
| Weight decay | $1 \times 10^{-4}$ |
| Gradient norm clip | 1.0 |

## B  Experiment Details

### B.1  VICON Model Details

Here we provide key architectural parameters of VICON model implementation in Table 4.

### B.2  Training Details

We implement our method in PyTorch Paszke et al. (2019) and utilize data parallel training Li et al. (2020) across two NVIDIA RTX 4090 GPUs. We employ the AdamW optimizer with a cosine learning rate schedule that includes a linear warmup phase. We apply gradient clipping with a maximum norm of 1.0 to ensure training stability. All optimization parameters are detailed in Table 5.

### B.3  MPP Details

For a fair comparison, we retrain MPP McCabe et al. (2023) using our training dataset with the same model configurations and optimizer hyperparameters (batch size is set to the maximum possible on our device). We evaluate MPP using the same testing setup and metric.

---

**Algorithm 1:** GenerateSingleStepStrategy

---

**Input:** D: Number of demonstrations to use
R: Number of ground truth reference steps
T: Total steps in sequence
**Output:** S: Strategy list of example pairs and question pairs

**1** $S \leftarrow [\,]$
**2** $Ei \leftarrow range(0, D)$                             `/* example input indices: 0,1,...,D-1 */`
**3** $Eo \leftarrow range(1, D+1)$                      `/* example output indices: 1,2,...,D */`
**4** **for** *i from R to T-1* **do**
**5**      $Qi \leftarrow i - 1$                             `/* question input is previous frame */`
**6**      $Qo \leftarrow i$                                  `/* question output is current frame */`
**7**      S.append((Ei, Eo), Qi, Qo)
**8** **end**
**9** **return** S

---

## B.4 DPOT Details

For a fair comparison, we retrain DPOT Hao et al. (2024) using our training dataset with the same model configurations and optimizer hyperparameters (batch size is set to the maximum possible on our device). We evaluate DPOT using the same testing setup and metric.

## B.5 Specialist U-Net Details

We use U-Net-Mod64, the strongest baseline from the PDEArena benchmark Gupta & Brandstetter (2022), trained individually per dataset. This architecture employs a U-Net with modified residual blocks and 64 base channels. We keep the architecture consistent across all datasets, only adjusting the input/output channel dimensions to match each dataset's state variables. We believe this is fair because in realistic deployment, one cannot architecturally re-engineer a model for every new measurement stream encountered.

Additionally, we include instance normalization matching VICON's preprocessing (channel-wise normalization using statistics computed from initial frames), which we found essential for stabilizing training on the PDEBench datasets (PDEBench-Comp-LowVis and PDEBench-Comp-HighVis). Without this normalization, training on compressible flows diverges due to the large magnitude variations in the pressure channel. The batch size is set to the maximum possible on our device. Each specialist model is trained on its respective dataset only, in contrast to VICON's joint multi-physics training.

# C Algorithms and Examples of Strategy Generation

## C.1 Strategy with Full Temporal Sequence

**Algorithm for single-step strategy.** Algorithm 1 presents our approach for generating single-step rollout strategies when all temporal frames are available. This strategy maintains a fixed stride of 1 between consecutive frames, providing stable but potentially error-accumulating predictions.

**Algorithm for flexible-step strategy.** Algorithm 2 describes our approach for generating flexible-step strategies with variable strides up to a maximum value. This approach reduces the total number of rollout steps required, potentially mitigating error accumulation for long sequences.

**Rollout Strategy Example.** We demonstrate two rollout strategies in Tables 6 and 7, showing single-step and flexible-step strategies, respectively. In both cases, the initial frames span from time step 0 to 9, and we aim to predict the trajectory up to time step 20.

---

**Algorithm 2:** GenerateFlexibleStepStrategy

---

**Input:** D: Number of demonstrations to use
R: Number of ground truth reference steps
M: Maximum stride between pairs
T: Total steps in sequence
**Output:** S: Strategy list of example pairs and question pairs
```
/* Require R >= M + 1 to ensure examples exist for each stride             */
```
**1** S ← [ ]
**2** Ec ← {}                                          /* example condition indices by stride */
**3** Eq ← {}                                             /* example query indices by stride */
```
/* Prepare example pairs for each stride                                   */
```
**4** **for** *s from 1 to M* **do**
**5**    Nd ← min(D, R - s)                            /* available examples for this stride */
**6**    Eqo ← range(R - Nd, R)                              /* output indices for examples */
```
    /* If we need more examples than available, repeat them                */
```
**7**    reps ← ⌈D / Nd⌉                                      /* ceiling division */
**8**    Eqo ← repeat(Eqo, reps)[0:D]                    /* repeat and truncate to D */
**9**    Eqo.sort()                                         /* ensure increasing order */
**10**    Ec[s] ← Eqo - s                         /* input indices are s steps before outputs */
**11**    Eq[s] ← Eqo                                         /* store output indices */
**12** **end**
```
/* Generate rollout strategy                                               */
```
**13** **for** *i from R to T-1* **do**
**14**    dist ← i - R + 1                              /* distance from last reference frame */
**15**    s ← min(dist, M)                                  /* select appropriate stride */
**16**    Qi ← i - s                                        /* question input index */
**17**    Qo ← i                                            /* question output index */
**18**    S.append((Ec[s], Eq[s]), Qi, Qo)
**19** **end**
**20** **return** S

---

Table 6: Single-step Rollout Strategy Example

| Rollout index | Examples (COND, QOI) | Question COND | Predict QOI |
|---|---|---|---|
| 1 | (0,1) (1,2) (2,3) (3,4) (4,5) (5,6) (6,7) (7,8) (8,9) | 9 | 10 |
| 2 | (0,1) (1,2) (2,3) (3,4) (4,5) (5,6) (6,7) (7,8) (8,9) | 10 | 11 |
| 3 | (0,1) (1,2) (2,3) (3,4) (4,5) (5,6) (6,7) (7,8) (8,9) | 11 | 12 |
| 4 | (0,1) (1,2) (2,3) (3,4) (4,5) (5,6) (6,7) (7,8) (8,9) | 12 | 13 |
| 5 | (0,1) (1,2) (2,3) (3,4) (4,5) (5,6) (6,7) (7,8) (8,9) | 13 | 14 |
| 6 | (0,1) (1,2) (2,3) (3,4) (4,5) (5,6) (6,7) (7,8) (8,9) | 14 | 15 |
| 7 | (0,1) (1,2) (2,3) (3,4) (4,5) (5,6) (6,7) (7,8) (8,9) | 15 | 16 |
| 8 | (0,1) (1,2) (2,3) (3,4) (4,5) (5,6) (6,7) (7,8) (8,9) | 16 | 17 |
| 9 | (0,1) (1,2) (2,3) (3,4) (4,5) (5,6) (6,7) (7,8) (8,9) | 17 | 18 |
| 10 | (0,1) (1,2) (2,3) (3,4) (4,5) (5,6) (6,7) (7,8) (8,9) | 18 | 19 |
| 11 | (0,1) (1,2) (2,3) (3,4) (4,5) (5,6) (6,7) (7,8) (8,9) | 19 | 20 |

As shown in Table 7, the flexible-step strategy initially uses smaller strides to build sufficient examples, then employs maximum strides for later rollouts, as detailed in Section 4.5. We note that repeated in-context examples appear in Table 7, which is common when the maximum stride is large and the initial frames cannot form enough examples. While the number of in-context examples in VICON is flexible and our model can

Table 7: Flexible-step Rollout Strategy Example ($s_{\max} = 5$)

| Rollout index | Examples (COND, QOI) | Question COND | Predict QOI |
|---|---|---|---|
| 1 | (0,1) (1,2) (2,3) (3,4) (4,5) (5,6) (6,7) (7,8) (8,9) | 9 | 10 |
| 2 | (0,2) (0,2) (1,3) (2,4) (3,5) (4,6) (5,7) (6,8) (7,9) | 9 | 11 |
| 3 | (0,3) (0,3) (1,4) (1,4) (2,5) (3,6) (4,7) (5,8) (6,9) | 9 | 12 |
| 4 | (0,4) (0,4) (1,5) (1,5) (2,6) (2,6) (3,7) (4,8) (5,9) | 9 | 13 |
| 5 | (0,5) (0,5) (1,6) (1,6) (2,7) (2,7) (3,8) (3,8) (4,9) | 9 | 14 |
| 6 | (0,5) (0,5) (1,6) (1,6) (2,7) (2,7) (3,8) (3,8) (4,9) | 10 | 15 |
| 7 | (0,5) (0,5) (1,6) (1,6) (2,7) (2,7) (3,8) (3,8) (4,9) | 11 | 16 |
| 8 | (0,5) (0,5) (1,6) (1,6) (2,7) (2,7) (3,8) (3,8) (4,9) | 12 | 17 |
| 9 | (0,5) (0,5) (1,6) (1,6) (2,7) (2,7) (3,8) (3,8) (4,9) | 13 | 18 |
| 10 | (0,5) (0,5) (1,6) (1,6) (2,7) (2,7) (3,8) (3,8) (4,9) | 14 | 19 |
| 11 | (0,5) (0,5) (1,6) (1,6) (2,7) (2,7) (3,8) (3,8) (4,9) | 15 | 20 |

---

**Algorithm 3:** GetAvailablePairs

**Input:** D: Number of demonstrations needed
dt: Desired time stride between pairs
Fa: List of indices of available frames
**Output:** P: List of input-output index pairs with the specified stride

```
1  Fa.sort()
2  P ← [ ]                                              /* Initialize empty pairs list */
   /* Find all pairs with stride dt                                                   */
3  for i from 0 to len(Fa) - 1 do
4      for j from i + 1 to len(Fa) - 1 do
5          if Fa[j] - Fa[i] == dt then
6              P.append((Fa[i], Fa[j]))
7          end
8      end
9  end
10 if P is empty then
11     return [ ]
12 end
   /* If we don't have enough unique pairs, repeat them                               */
13 if len(P) < D then
14     reps ← ⌈D / len(P)⌉                                        /* ceiling division */
15     RP ← repeat(P, reps)[0:D]                    /* repeat pairs and truncate to D */
16     P ← RP
17 end
18 return P
```

---

accommodate fewer examples than the designed length, our preliminary experiments indicate that the model performs better with more in-context examples, even when some examples are repeated.

## C.2 Strategy with Imperfect Temporal Sequence

When facing imperfect temporal sampling where certain frames are missing, we can still form valid demonstration pairs from the available frames. Algorithm 3 presents this adaptive pair selection process, which serves as a fundamental building block for all strategy generation algorithms in scenarios with irregular temporal data.

---

**Algorithm 4:** GenerateSingleStepStrategyWithDrops

---

**Input:** D: Number of demonstrations to use
S: Fixed stride between pairs
T: Total steps in sequence
Fa: List of indices of available frames
**Output:** St: Strategy list of (E, Qi, Qo) tuples

**1** St ← [ ]
**2** Fa.sort()
**3** Fs ← Fa[-1]                                             /* the starting frame for rollout */
**4** P ← GetAvailablePairs(D, S, Fa)                              /* pairs with fixed stride */
**5** **if** *P is empty* **then**
**6** | **return** [ ]                               /* No available pairs with the specified stride */
**7** **end**
**8** Fc ← Fa.copy()                              /* accumulated frames (current + predicted) */
**9** **for** *i from (Fs + 1) to (T - 1)* **do**
**10** | Ci ← i - S                                              /* potential condition index */
**11** | **if** *Ci ∉ Fc* **then**
**12** | | **continue**                               /* Cannot predict frame i with stride S */
**13** | **end**
**14** | Pdt ← P[s]
| |                                          /* The example pairs for predicting frame i */
**15** | Qi ← i - s                                              /* question input index */
**16** | Qo ← i                                                 /* question output index */
**17** | S.append(Pdt, Qi, Qo)
**18** | Fc.append(Qo)                         /* The predicted frame can be used for future steps */
**19** **end**
**20** **return** St

---

**Algorithm for single-step strategy with drops.** Algorithm 4 extends the single-step strategy to handle scenarios where frames are missing from the input sequence, adaptively forming strategies from available frames while maintaining the fixed stride constraint.

**Algorithm for flexible-step strategy.** Algorithm 5 presents our solution for generating flexible-step rollout strategies when frames are missing, dynamically selecting appropriate strides based on available frames and previous predictions.

**Rollout Strategy Example with Missing Frames.** Tables 8 and 9 illustrate our adaptive strategies when frames 2, 5, and 9 are missing from the initial sequence (with all other settings identical to those in Appendix C.1). The tables demonstrate single-step and flexible-step (max stride: 3) approaches, respectively, highlighting how our algorithm dynamically selects appropriate example pairs to maintain prediction capability despite missing temporal data.

## D  Additional Experimental Results.

### D.1  Ablation Studies

**Impact of Patch Resolutions.** We conducted ablation studies on patch resolution by varying patch sizes (4, 8, 16, 32, 64) to find a balance between spatial granularity and computational resource constraint. While smaller patches theoretically capture finer details, they generate longer token sequences, hitting memory caps due to transformer's quadratic complexity. For patch size 8, we reduced token dimensions (512→256) and feedforward dimensions (1024→512). For patch size 4, we applied further reductions (token dim 128, feedforward dim 256, layers 10→5) to fit within 80GB A100 memory—yet training still required 188 hours (vs

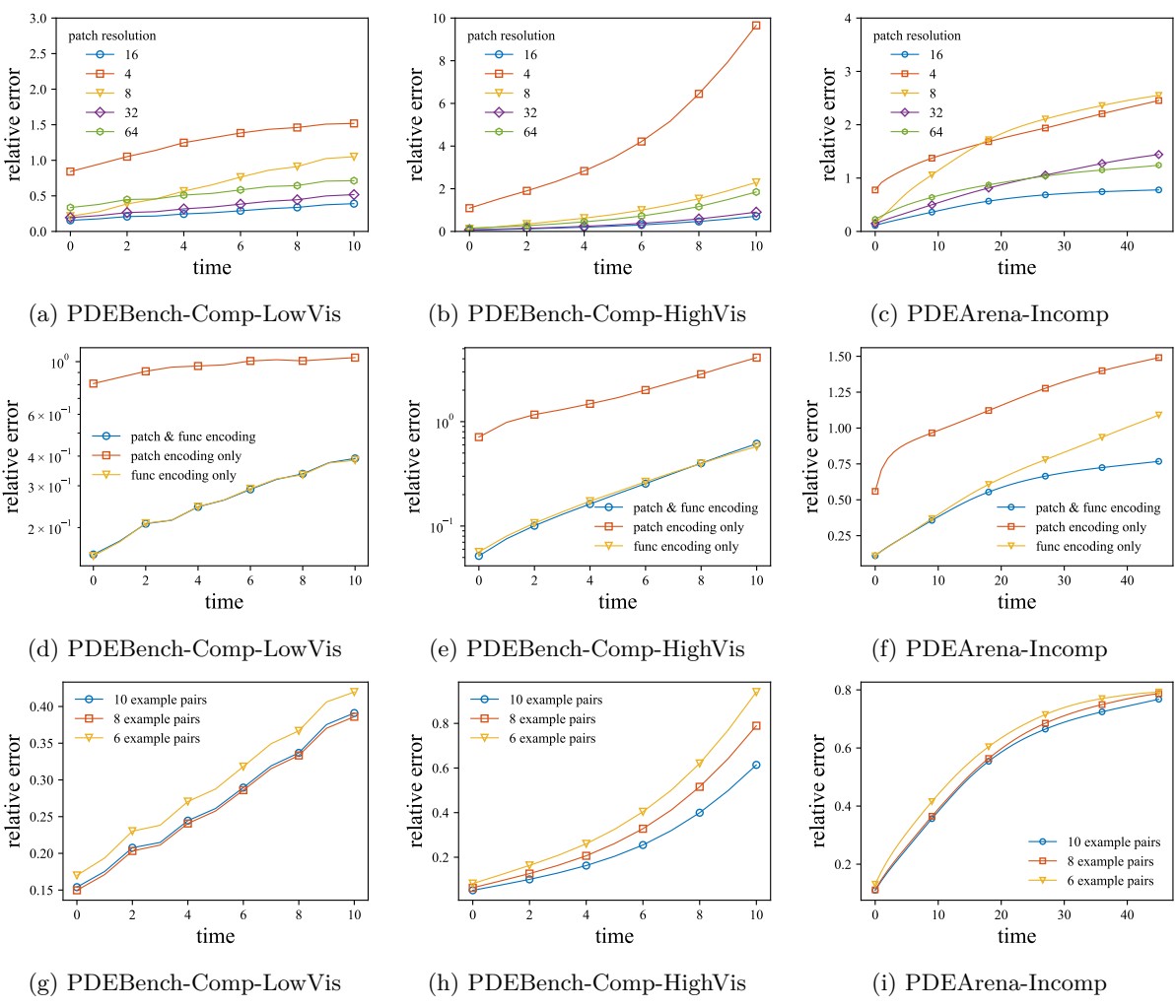

Figure 4: Ablation studies across the three datasets. **Top row (a-c):** Impact of patch resolutions $(4, 8, 16, 32, 64)$ showing optimal performance at patch size 16. **Middle row (d-f):** Effect of different positional encoding combinations. **Bottom row (g-i):** Performance variation with different context lengths $(6, 8, 10$ pairs$)$.

---

**Algorithm 5:** GenerateFlexibleStepStrategyWithDrops

---

**Input:** D: Number of demonstrations to use
M: Maximum stride between pairs for rollout
T: Total steps in the sequence
Fa: List of indices of available frames
**Output:** S: Strategy list of (E, Qi, Qo) tuples, where E are the list of example pair indices (Ei, Eo) and
Qi, Qo are the question and output indices

**1** S ← [ ]
**2** Fa.sort()
**3** Fs ← Fa[-1]                                                    /* the starting frame for rollout */
**4** P ← {}                                                        /* dictionary of available pairs */
**5** **for** *dt from 1 to M* **do**
**6**     Pdt ← GetAvailablePairs(D, dt, Fa)                           /* pairs with stride dt in Fa */
**7**     **if** *Pdt is not empty* **then**
**8**        P[dt] ← Pdt
**9**     **end**
**10** **end**
**11** Ms ← max(P.keys())                                             /* maximum available stride */
**12** Fc ← Fa.copy()                                    /* accumulated frames (current + predicted) */
**13** **for** *i from (Fs + 1) to (T - 1)* **do**
**14**     dt ← i - Fs
**15**     Mt ← min(dt, M, Ms)                                    /* maximum stride for current step */
**16**     found ← False
**17**     **for** *s in P.keys().sorted(reverse=True) where s ≤ Mt* **do**
**18**        Ci ← i - s                                        /* potential condition index */
**19**        **if** *Ci ∈ Fc* **then**
          /* Found a starting index to obtain frame i                              */
**20**           found ← True
**21**           **break**
**22**        **end**
**23**     **end**
**24**     **if** *not found* **then**
       /* We cannot predict frame i                                            */
**25**        **continue**
**26**     **end**
    /* The example pairs for predicting frame i                             */
**27**     Pdt ← P[s]
**28**     Qi ← i - s                                           /* question input index */
**29**     Qo ← i                                               /* question output index */
**30**     S.append(Pdt, Qi, Qo)
**31**     Fc.append(Qo)
          /* The Qo frame is obtained and can be used for future rollouts */
**32** **end**
**33** **return** S

---

58 hours for ps=16) and inference slowed to 9.55 sec/step (vs 8.7 ms for ps=16). Patches below $4 \times 4$ remained infeasible even with these aggressive reductions. Figure 4(a-c) shows that patch size 16 achieves optimal performance across all datasets—balancing sequence length and representational capacity. Performance degradation with coarser patches (32, 64) aligns with expectations, while degradation with finer patches stems from the necessary reduction in hidden dimensions, highlighting a fundamental challenge in scaling to 3D applications.

Table 8: Single-step Rollout Strategy Example with Missing Frames

| Rollout index | Examples (COND, QOI) | Question COND | Predict QOI |
|---|---|---|---|
| 1 | (0,1) (0,1) (0,1) (3,4) (3,4) (6,7) (6,7) (7,8) (7,8) | 8 | 9 |
| 2 | (0,1) (0,1) (0,1) (3,4) (3,4) (6,7) (6,7) (7,8) (7,8) | 9 | 10 |
| 3 | (0,1) (0,1) (0,1) (3,4) (3,4) (6,7) (6,7) (7,8) (7,8) | 10 | 11 |
| 4 | (0,1) (0,1) (0,1) (3,4) (3,4) (6,7) (6,7) (7,8) (7,8) | 11 | 12 |
| 5 | (0,1) (0,1) (0,1) (3,4) (3,4) (6,7) (6,7) (7,8) (7,8) | 12 | 13 |
| 6 | (0,1) (0,1) (0,1) (3,4) (3,4) (6,7) (6,7) (7,8) (7,8) | 13 | 14 |
| 7 | (0,1) (0,1) (0,1) (3,4) (3,4) (6,7) (6,7) (7,8) (7,8) | 14 | 15 |
| 8 | (0,1) (0,1) (0,1) (3,4) (3,4) (6,7) (6,7) (7,8) (7,8) | 15 | 16 |
| 9 | (0,1) (0,1) (0,1) (3,4) (3,4) (6,7) (6,7) (7,8) (7,8) | 16 | 17 |
| 10 | (0,1) (0,1) (0,1) (3,4) (3,4) (6,7) (6,7) (7,8) (7,8) | 17 | 18 |
| 11 | (0,1) (0,1) (0,1) (3,4) (3,4) (6,7) (6,7) (7,8) (7,8) | 18 | 19 |

Table 9: Flexible-step Rollout Strategy Example with Missing Frames (max stride: 3)

| Rollout index | Examples (COND, QOI) | Question COND | Predict QOI |
|---|---|---|---|
| 1 | (0,1) (0,1) (0,1) (3,4) (3,4) (6,7) (6,7) (7,8) (7,8) | 8 | 9 |
| 2 | (1,3) (1,3) (1,3) (4,6) (4,6) (4,6) (6,8) (6,8) (6,8) | 8 | 10 |
| 3 | (0,3) (0,3) (0,3) (1,4) (1,4) (3,6) (3,6) (4,7) (4,7) | 8 | 11 |
| 4 | (0,3) (0,3) (0,3) (1,4) (1,4) (3,6) (3,6) (4,7) (4,7) | 9 | 12 |
| 5 | (0,3) (0,3) (0,3) (1,4) (1,4) (3,6) (3,6) (4,7) (4,7) | 10 | 13 |
| 6 | (0,3) (0,3) (0,3) (1,4) (1,4) (3,6) (3,6) (4,7) (4,7) | 11 | 14 |
| 7 | (0,3) (0,3) (0,3) (1,4) (1,4) (3,6) (3,6) (4,7) (4,7) | 12 | 15 |
| 8 | (0,3) (0,3) (0,3) (1,4) (1,4) (3,6) (3,6) (4,7) (4,7) | 13 | 16 |
| 9 | (0,3) (0,3) (0,3) (1,4) (1,4) (3,6) (3,6) (4,7) (4,7) | 14 | 17 |
| 10 | (0,3) (0,3) (0,3) (1,4) (1,4) (3,6) (3,6) (4,7) (4,7) | 15 | 18 |
| 11 | (0,3) (0,3) (0,3) (1,4) (1,4) (3,6) (3,6) (4,7) (4,7) | 16 | 19 |

**Positional Encodings.** We evaluated different combinations of positional encodings as shown in Figure 4(d-f). Our architecture employs two types of encodings: patch position encodings (for spatial relationships between patches) and function encodings (differentiating input/output in different pairs) (Section 4.2).

The results demonstrate that including both encoding types consistently yields the best performance across all datasets. Notably, function encodings have a more significant impact than patch encodings, highlighting the importance of distinguishing between the condition and qoi beyond what causal masking (equation 6) provides.

The impact of patch encodings varies across datasets: they show minimal effect on the PDEBench datasets (PDEBench-Comp-LowVis and PDEBench-Comp-HighVis) yet significantly improve performance on PDEArena-Incomp. We attribute this to the different stiffness of these systems. In compressible flows (PDEBench), spatial interactions naturally decay with distance, creating consistent and monotonic spatial correlations between patches. In contrast, Navier-Stokes equations exhibit infinite stiffness where correlations between any spatial locations are instantaneous and determined by the global velocity field, creating non-monotonic and case-dependent relationships. Explicitly encoding spatial positions therefore becomes particularly beneficial for PDEArena-Incomp, as it reduces the learning complexity for these non-local dependencies.

**Varying Context Length.** We investigated how the number of in-context examples affects model performance, as shown in Figure 4(g-i). Following insights from the original ICON work Yang et al. (2023a), we evaluated context lengths of 6, 8, and 10 pairs, balancing performance against computational efficiency. The results show that 10 in-context examples deliver significantly better results for PDEBench-Comp-LowVis compared to shorter contexts, while for the remaining datasets, 10 and 8 in-context examples perform similarly (both outperforming 6 examples).

**Alternative Architecture: CNN-ICON.** We explored an alternative architecture (CNN-ICON) that uses a CNN encoder to compress entire frames into single latent vectors, followed by the ICON transformer structure. The encoder consists of a 4-level CNN with residual blocks ($7 \times 128 \times 128 \rightarrow 512$-dim latent via global average pooling), paired with a symmetric CNN decoder. This results in 1 token per frame compared to VICON's 64 tokens ($8 \times 8$ patches). We matched the model sizes: 86.5M parameters (vs VICON's 87.8M), identical transformer configuration (6 layers, 8 heads, dim=512), and same training protocol (200K steps).

Table 10: CNN-ICON vs VICON comparison (normalized RMSE, scaled by std)

| Dataset | Model | Step 1 | Step 5 | Step 10 | Last | Avg |
|---|---|---|---|---|---|---|
| PDEArena-Incomp | CNN-ICON | 0.323 | 0.467 | 0.643 | 1.194 | 0.912 |
| | VICON | 0.110 | 0.206 | 0.305 | 0.680 | 0.485 |
| PDEBench-Comp-LowVis | CNN-ICON | 0.750 | 0.875 | 1.012 | 1.013 | 0.905 |
| | VICON | 0.154 | 0.245 | 0.375 | 0.391 | 0.271 |
| PDEBench-Comp-HighVis | CNN-ICON | 0.390 | 0.767 | 1.939 | 2.364 | 1.162 |
| | VICON | 0.052 | 0.163 | 0.498 | 0.614 | 0.301 |

As shown in Table 10, VICON outperforms CNN-ICON by 1.9–3.9× across all datasets. We attribute this performance gap to information loss when compressing entire frames into single vectors, which discards spatial structure crucial for accurate PDE prediction. This finding motivated our patch-based design, which preserves local spatial information while maintaining computational efficiency.

## D.2 Analysis of Checkerboard Artifacts in PDEBench-Comp-HighVis

In our experiments, we observed block-like artifacts in the error/difference maps for the PDEBench-Comp-HighVis dataset that align with the model's $16 \times 16$ patch structure. Through detailed analysis, we identified two contributing factors: 1) drastic field value variation across the trajectory time horizon, and 2) a training-inference normalization mismatch.

**Drastic field value variation across trajectory time horizon.** High-viscosity flows rapidly dissipate kinetic energy, causing velocity fields to become nearly stationary after approximately 10 frames. Our analysis of standard deviations across different time ranges reveals these differences:

Table 11: Standard deviation analysis across time phases for PDEBench datasets. The bolded velocity channels show PDEBench-Comp-HighVis has extreme variation between early and late phases, unlike PDEBench-Comp-LowVis with nearly consistent variation. This dynamic difference makes training a uniformed model particularly challenging.

| Dataset | Velocity u | | | Velocity v | | | Pressure |
|---|---|---|---|---|---|---|---|
| | All | 1-10 | >10 | All | 1-10 | >10 | All |
| PDEBench-Comp-HighVis | 0.359 | 0.511 | **0.091** | 0.351 | 0.506 | **0.049** | 14.254 |
| PDEBench-Comp-LowVis | 0.989 | 1.104 | 0.873 | 1.036 | 1.165 | 0.903 | 29.687 |

**Training-inference normalization mismatch.** During training, context pairs are sampled uniformly across all frames. During inference, context pairs are sampled from initial frames only. As shown in Table 11, the standard deviations at these two different time ranges are vastly different, which results in normalization mismatch and exacerbates prediction errors in the already-challenging problem. Future work could explore using global statistics from multiple trajectories instead of single-trajectory instance normalization during deployment. Other potential remedies include using overlapping patches or moving patch positions during rollout, though these represent engineering improvements orthogonal to our core contributions.

### D.3 Cross-Trajectory Context Sampling

We investigate whether VICON can leverage context pairs from different trajectories governed by the same underlying operator, as suggested in the original ICON framework. Using the PDEArena-Incomp dataset (which provides 32 distinct trajectories per operator), we compare VICON trained with intra-trajectory context only versus a mixture of intra- and cross-trajectory context (50/50 split).

Table 12: Cross-trajectory context experiment on PDEArena-Incomp (normalized RMSE, scaled by std)

| Training Setup | Inference Setup | Avg Error |
|---|---|---|
| VICON (trained with intra-traj only) | intra-traj context | 0.563 |
| | cross-traj context | 0.609 |
| VICON (trained with mixed context) | intra-traj context | 0.556 |
| | cross-traj context | 0.568 |

As shown in Table 12, VICON is capable of leveraging cross-trajectory information when trained appropriately. The model trained with mixed context achieves competitive performance in both settings, with only a marginal gap (0.556 vs 0.568) between intra- and cross-trajectory inference.

While these results demonstrate VICON's flexibility, we maintain that intra-trajectory context is more practical for deployment scenarios. Cross-trajectory sampling requires: (1) a pre-existing database of trajectories, and (2) a retrieval mechanism to ensure sampled trajectories share the same (often unknown) operator. In contrast, our intra-trajectory approach is self-contained and better suited for real-time forecasting where only the current trajectory is available.

### D.4 More Results and Visualizations

**Detailed Results.** Tables 13 and 14 summarize the relative and absolute $L^2$ rollout errors across different timesteps for all evaluated models on the three datasets.

Table 13: **(Comparison) Summary of Rollout Relative $L^2$ Error (scale by std)** for different methods across various cases. The best results are highlighted in bold.

| Rollout Relative $L^2$ Error | Case | Ours (single step) | Ours (flexible step) | DPOT | MPP |
|---|---|---|---|---|---|
| Step 1 [1e-2] | PDEArena-Incomp | 11.01 | 11.01 | **5.97** | 6.17 |
| | PDEBench-Comp-LowVis | **15.40** | **15.40** | 25.24 | 16.37 |
| | PDEBench-Comp-HighVis | **5.18** | **5.18** | 42.76 | 20.90 |
| Step 5 [1e-2] | PDEArena-Incomp | 22.68 | 20.62 | **11.70** | 14.81 |
| | PDEBench-Comp-LowVis | **24.45** | **24.45** | 36.05 | 45.45 |
| | PDEBench-Comp-HighVis | **16.25** | **16.25** | 286.3 | 42.93 |
| Step 10 [1e-2] | PDEArena-Incomp | 35.67 | 30.53 | **20.74** | 27.89 |
| | PDEBench-Comp-LowVis | **37.54** | **37.54** | 49.47 | 64.11 |
| | PDEBench-Comp-HighVis | **49.83** | **49.83** | 2016 | 144.46 |
| Last step [1e-2] | PDEArena-Incomp | 76.77 | 68.03 | **65.27** | 93.52 |
| | PDEBench-Comp-LowVis | **39.11** | **39.11** | 48.92 | 65.32 |
| | PDEBench-Comp-HighVis | **61.41** | **61.41** | 2866 | 185.3 |
| All average [1e-2] | PDEArena-Incomp | 56.27 | 48.50 | **41.20** | 55.95 |
| | PDEBench-Comp-LowVis | **27.08** | **27.08** | 37.72 | 46.68 |
| | PDEBench-Comp-HighVis | **30.06** | **30.06** | 821.9 | 72.37 |

Table 14: **(Comparison) Summary of Rollout Absolute $L^2$ Error** for different methods across various cases. The best results are highlighted in bold.

| Rollout $L^2$ Error | Case | Ours (single step) | Ours (flexible step) | DPOT | MPP |
|---|---|---|---|---|---|
| Step 1 [1e-2] | PDEArena-Incomp | 5.63 | 5.63 | **3.02** | 3.12 |
| | PDEBench-Comp-LowVis | **21.74** | **21.74** | 29.47 | 24.47 |
| | PDEBench-Comp-HighVis | **1.43** | **1.43** | 6.22 | 4.73 |
| Step 5 [1e-2] | PDEArena-Incomp | 10.38 | 9.46 | **5.37** | 6.79 |
| | PDEBench-Comp-LowVis | **31.23** | **31.23** | 43.57 | 57.50 |
| | PDEBench-Comp-HighVis | **2.34** | **2.34** | 14.48 | 6.52 |
| Step 10 [1e-2] | PDEArena-Incomp | 14.65 | 12.55 | **8.55** | 11.49 |
| | PDEBench-Comp-LowVis | **45.39** | **45.39** | 59.43 | 78.54 |
| | PDEBench-Comp-HighVis | **3.21** | **3.21** | 25.79 | 8.98 |
| Last Step [1e-2] | PDEArena-Incomp | 16.50 | 14.56 | **14.03** | 19.47 |
| | PDEBench-Comp-LowVis | **48.69** | **48.69** | 63.06 | 85.60 |
| | PDEBench-Comp-HighVis | **3.39** | **3.39** | 27.98 | 9.56 |
| All average [1e-2] | PDEArena-Incomp | 16.26 | 14.31 | **11.86** | 15.77 |
| | PDEBench-Comp-LowVis | **34.44** | **34.44** | 46.60 | 60.68 |
| | PDEBench-Comp-HighVis | **2.48** | **2.48** | 16.86 | 7.04 |

**Generalization to different timestep strides.** Figure 5 illustrates VICON's performance with varying timestep strides ($s_{\max} = 1, 3, 5, 7$) across all three datasets, demonstrating the model's ability to adapt to different temporal resolutions without retraining.

**Turbulence Kinetic Energy (TKE) Predictions.** Figure 6 displays the visualization of TKE fields for ground truth and model predictions, highlighting VICON's superior ability to preserve critical turbulent flow structures compared to baseline approaches.

**Comparison between joint training vs separate training.** Figure 7 and Table 15 quantify the performance differences between jointly trained and separately trained models, demonstrating the significant advantages of multi-physics training across all datasets.

**Computational Efficiency.** Table 16 summarizes the computational resources required by each method, showing VICON's advantages in terms of training cost, inference speed, and model parameter count compared to both DPOT and MPP baselines.

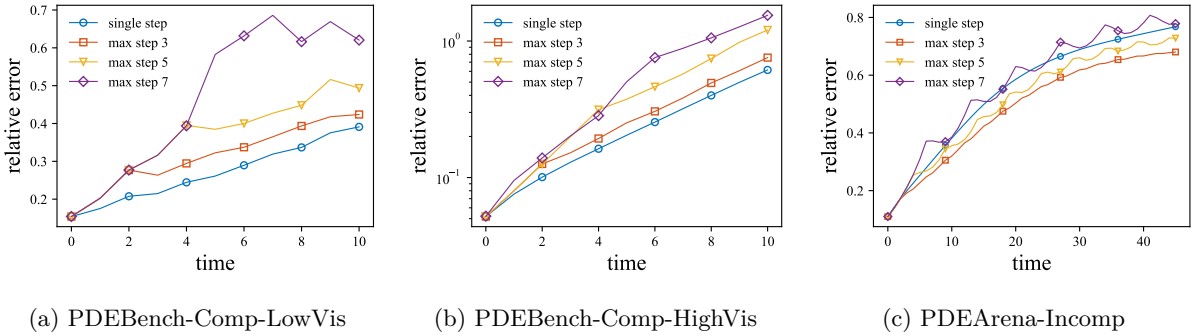

(a) PDEBench-Comp-LowVis      (b) PDEBench-Comp-HighVis      (c) PDEArena-Incomp

Figure 5: Comparison of rollout errors across different datasets, using single-step and flexible-step strategies with varying maximum step sizes ($s_{\max} = 1, 3, 5, 7$).

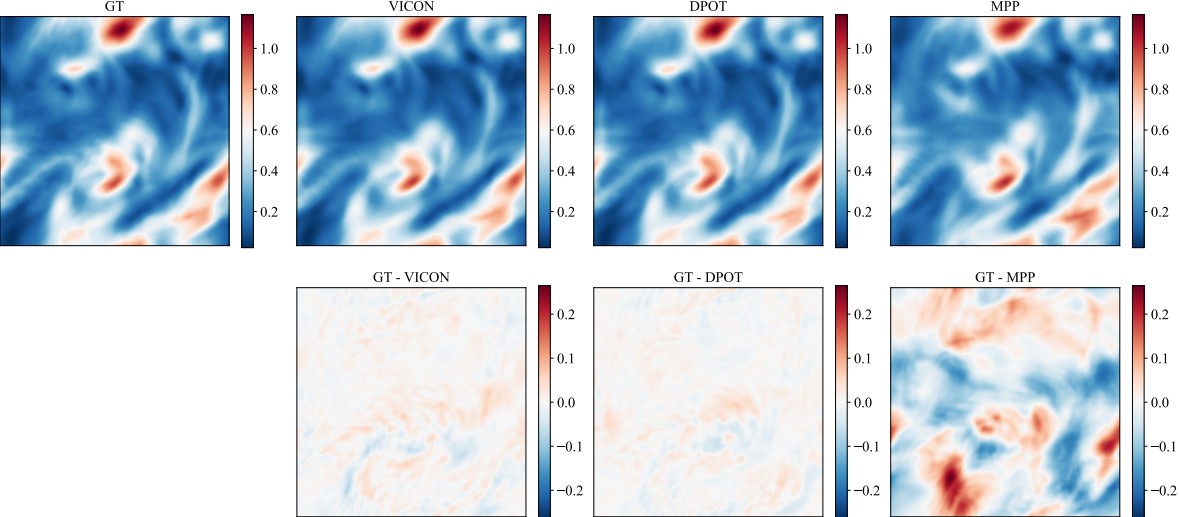

Figure 6: **Comparison of turbulence kinetic energy predictions.** (Top-left) Ground truth TKE field, (Top-right) model predictions, (Bottom) model error.

**Rollout with imperfect measurements.** Figure 8 show the rollout results with imperfect temporal measurements.

**Visualizations.** We compare the output of different models in Figure 9, Figure 10, and Figure 11. Figures 12 and 13 present additional visualizations of the VICON model outputs compared to ground truth and baseline predictions, highlighting the qualitative advantages of our approach.

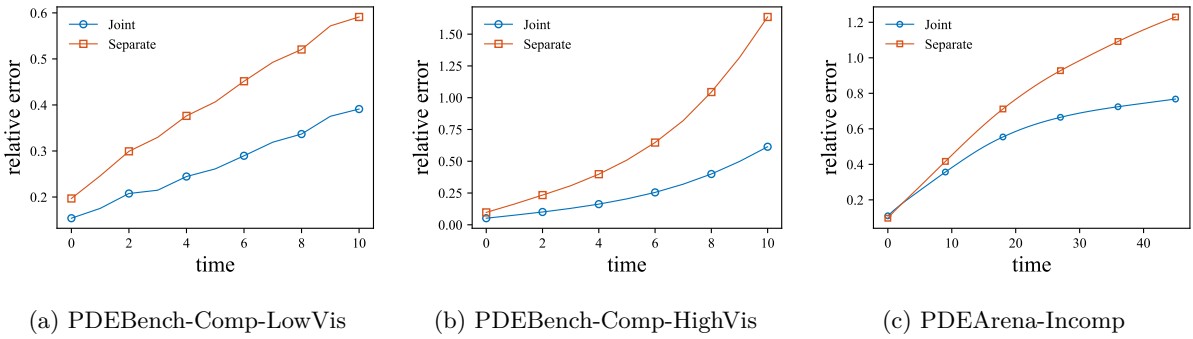

Figure 7: **Comparing rollout errors (single step, scale by std) for joint versus separate training strategies**. For separate training (individual models for each dataset), we maintain the same batch sizes as in joint training while adjusting training steps to be slightly more than one-third of the joint training duration, ensuring comparable total computational costs across both approaches.

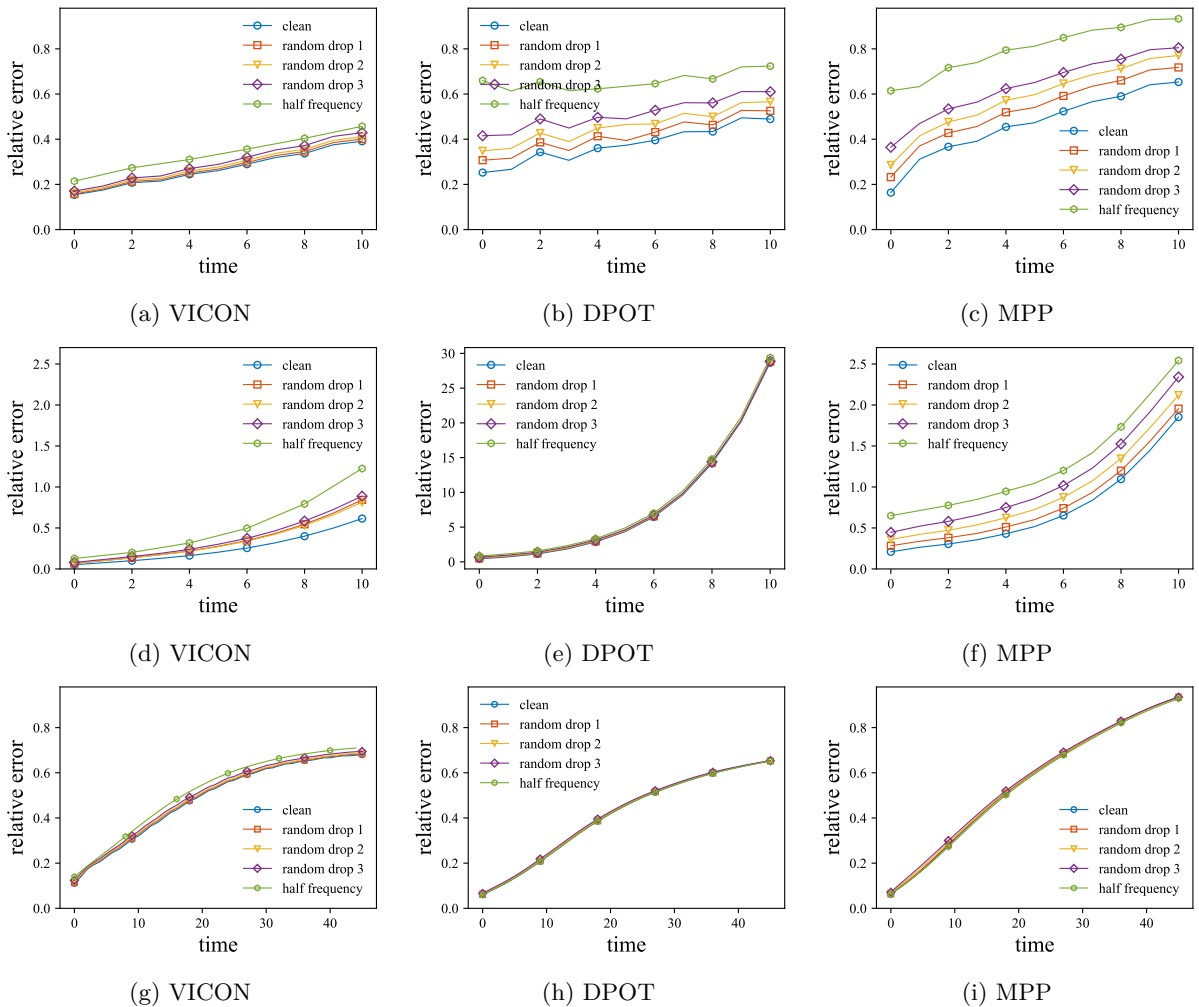

Figure 8: **Comparison of rollout errors (scale by std) with different input data noise levels across all datasets.** Each column (left to right) shows VICON, DPOT, and MPP models, while each row (top to bottom) represents PDEBench-Comp-LowVis, PDEBench-Comp-HighVis, and PDEArena-Incomp datasets. For MPP and DPOT which require fixed *dt* and context window, interpolation is used to generate missing frames, while VICON can directly handle irregular temporal data without interpolation.

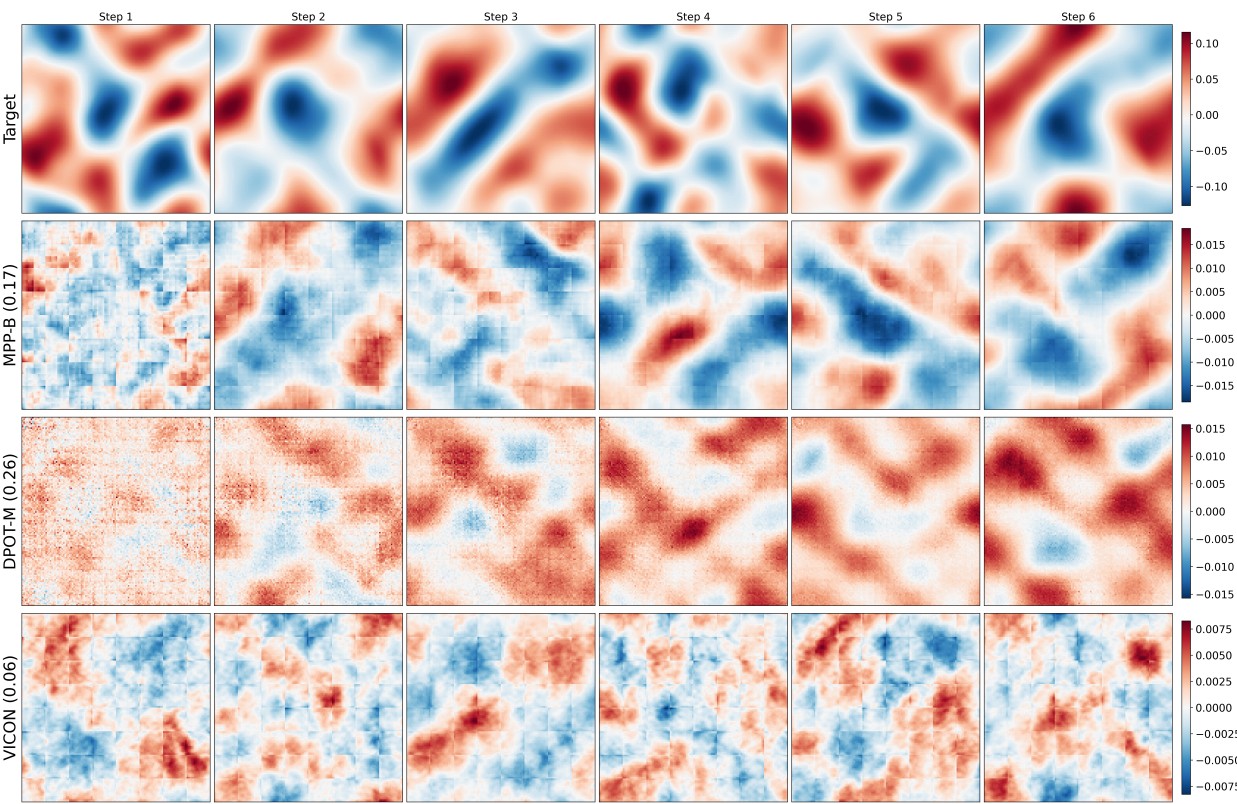

Figure 9: **Comparing outputs from different models.** The target is the first 6 output steps from PDEBench-Comp-HighVis dataset (x-velocity channel). For each model (each row), we display the difference between target and model output. Errors (rescaled by std) for the full trajectory are listed after the model names.

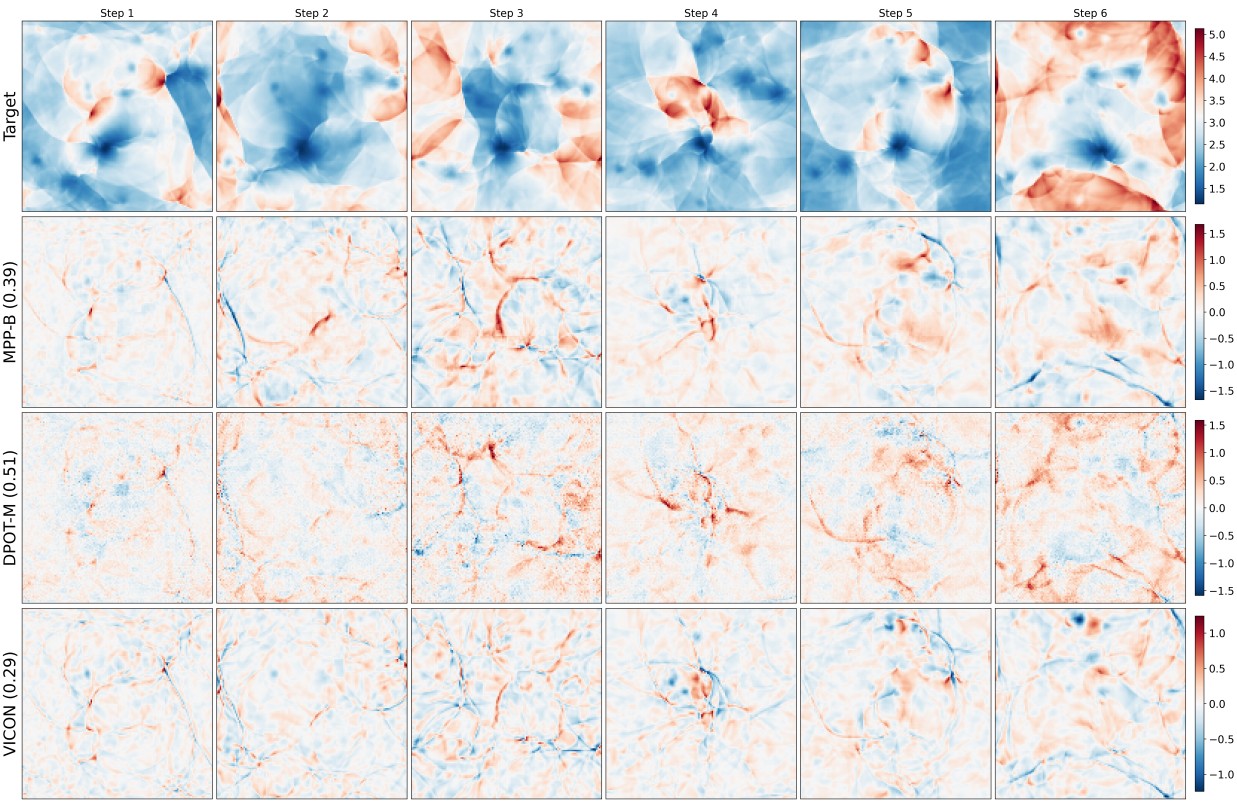

Figure 10: **Comparing outputs from different models.** The target is the first 6 output steps from PDEBench-Comp-LowVis dataset (pressure channel). For each model (each row), we display the difference between target and model output. Errors (rescaled by std) for the full trajectory are listed after the model names.

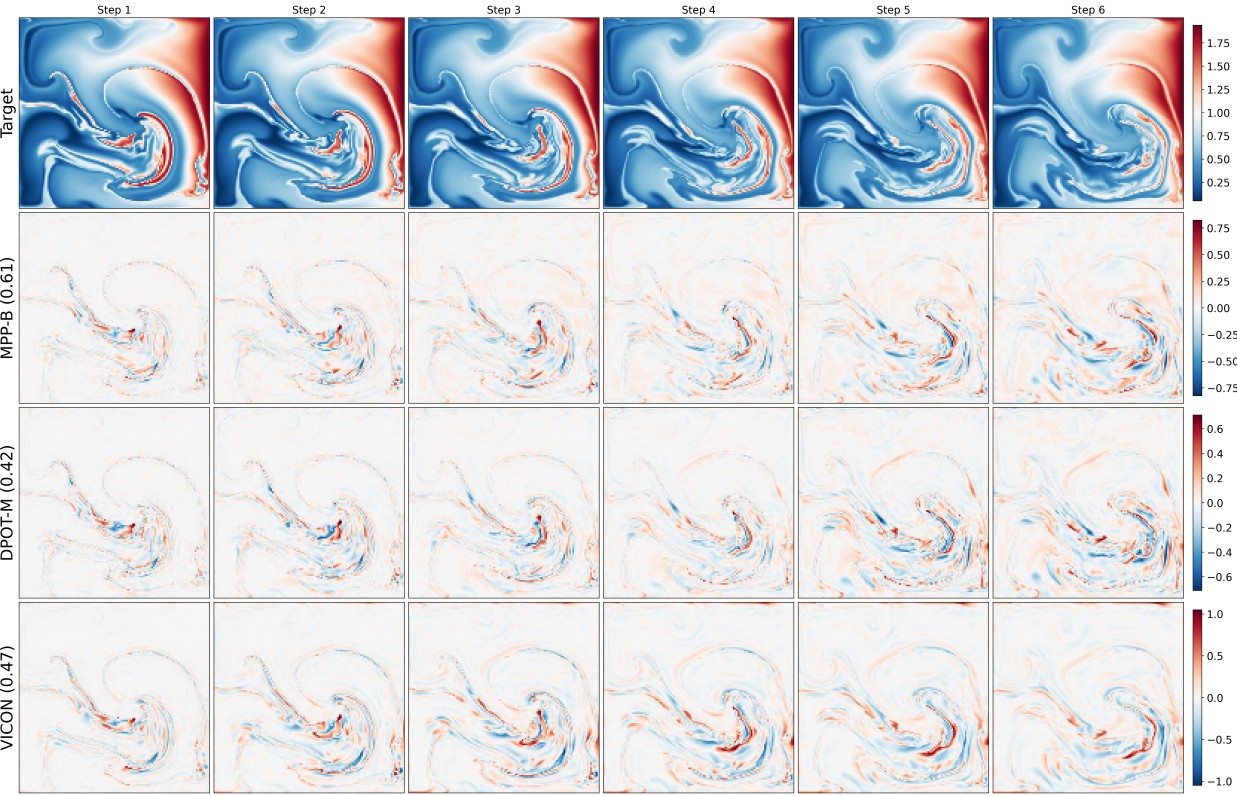

Figure 11: **Comparing outputs from different models.** The target is the first 6 output steps from PDEArena-Incomp dataset (particle density channel). For each model (each row), we display the difference between target and model output. Errors (rescaled by std) for the full trajectory are listed after the model names.

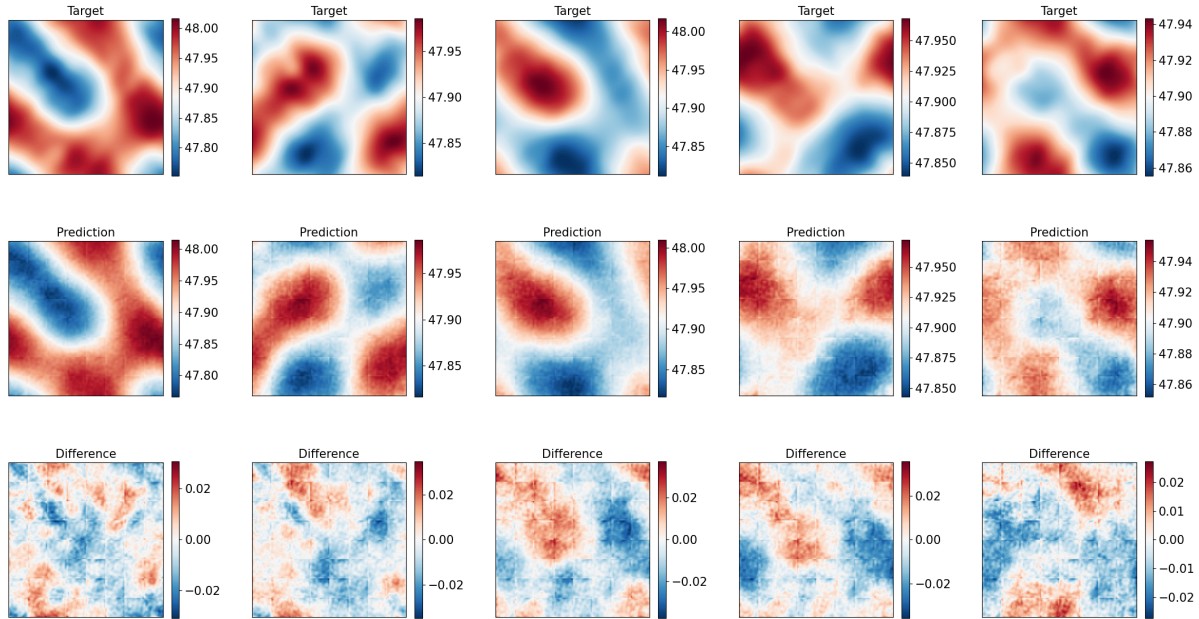

(a) A sequence of 5 output steps for PDEBench-Comp-HighVis dataset. The channel plotted is the pressure field in equation equation 12.

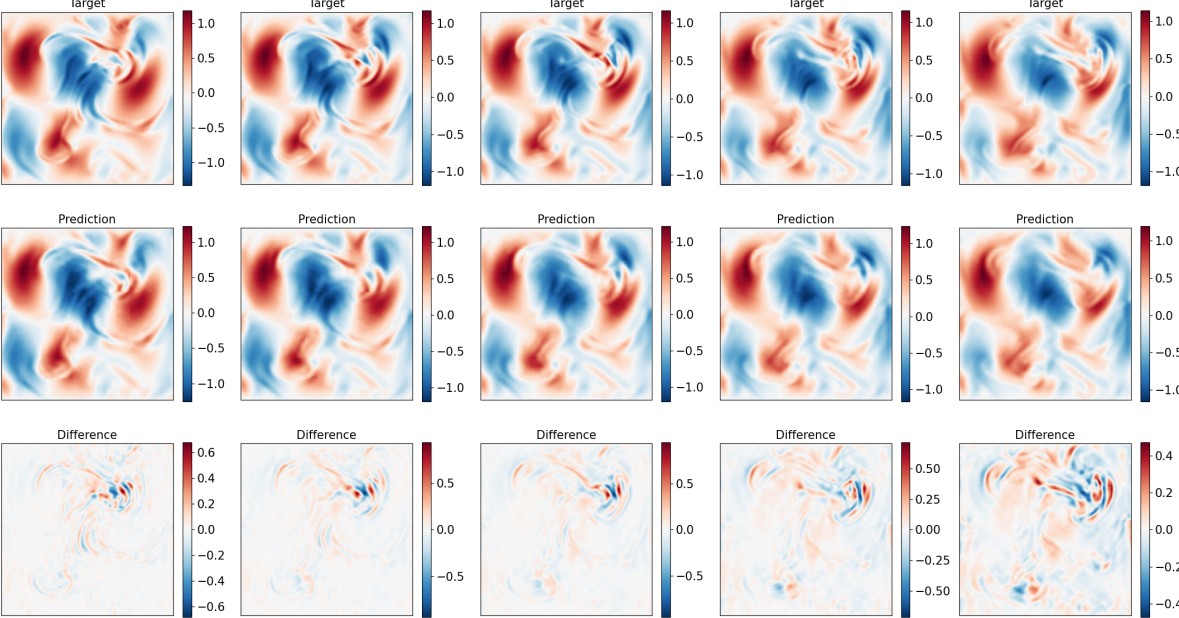

(b) A sequence of 5 output steps for PDEArena-Incomp dataset. The channel plotted is the x-velocity in equation equation 9.

Figure 12: **Example outputs for the VICON model.** (a) The pressure field of PDEBench-Comp-HighVis dataset, and (b) the x-velocity field of PDEArena-Incomp dataset. Each column represents a different timestep.

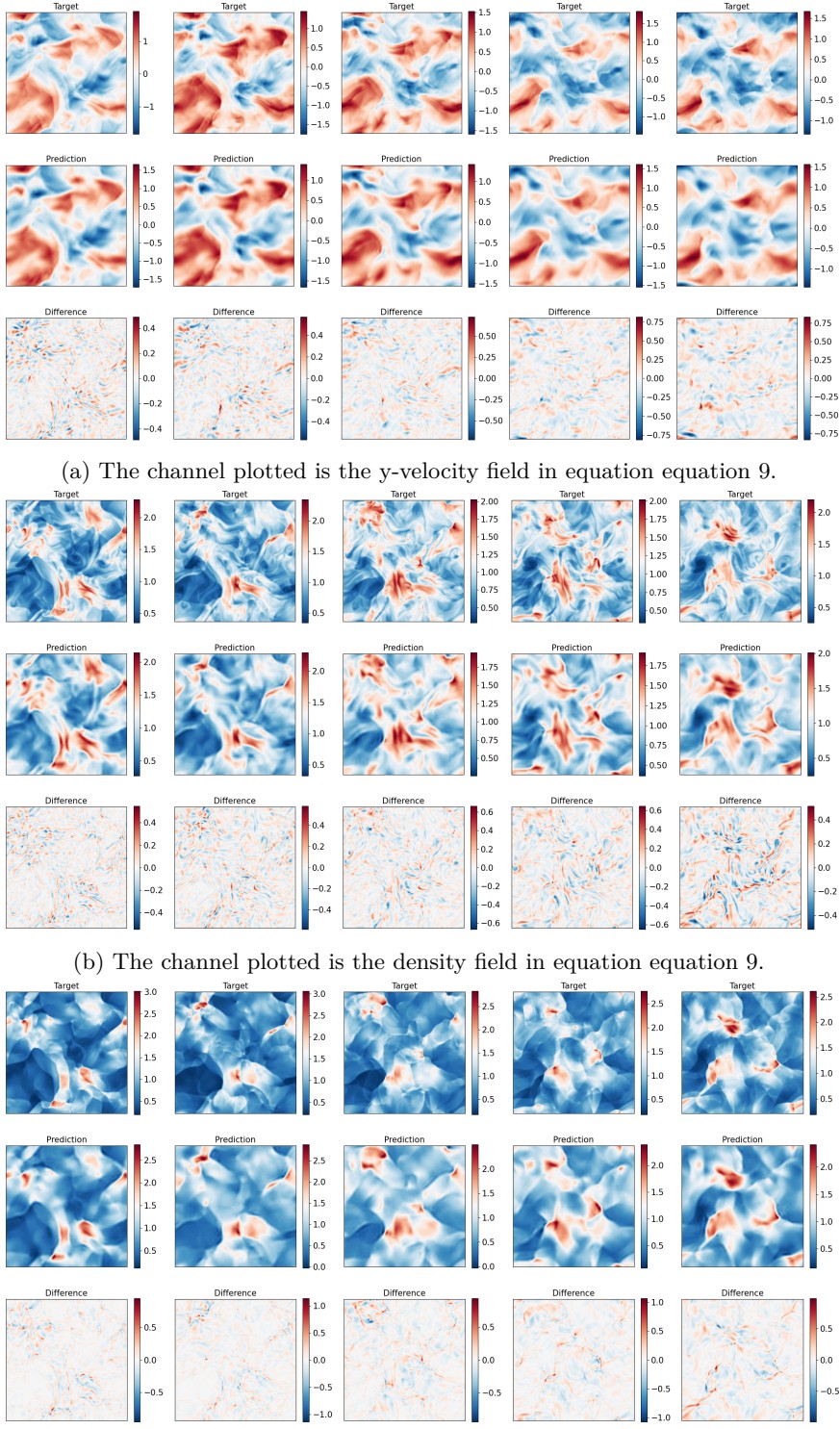

(a) The channel plotted is the y-velocity field in equation equation 9.

(b) The channel plotted is the density field in equation equation 9.

(c) The channel plotted is the pressure field in equation equation 9.

Figure 13: **More example outputs for the VICON model.** Showing 5 output steps for the PDEBench-Comp-LowVis dataset as governed by equation equation 9: (a) y-velocity, (b) density, and (c) pressure fields. Each column represents a different timestep.

Table 15: **Summary of Rollout Relative $L^2$ Error Metrics (single step, scale by std) for 2 Training Strategies**. The best results are highlighted in bold. For separate training (a model on each dataset), each run's batch size is controlled to be the same as joint training. To ensure fair comparison, we maintain the same batch sizes as joint training for each separate training run while adjusting their training steps to be slightly more than one-third of the joint training duration, ensuring comparable total computational costs.

| Rollout Relative $L^2$ Error [1e-2] | Case | Joint | Separate |
|---|---|---|---|
| | PDEArena-Incomp | 11.10 | **9.66** |
| Step 1 | PDEBench-Comp-LowVis | **15.61** | 19.68 |
| | PDEBench-Comp-HighVis | **5.79** | 9.74 |
| | PDEArena-Incomp | **23.00** | 24.09 |
| Step 5 | PDEBench-Comp-LowVis | **24.56** | 37.63 |
| | PDEBench-Comp-HighVis | **19.73** | 39.81 |
| | PDEArena-Incomp | **36.18** | 41.66 |
| Step 10 | PDEBench-Comp-LowVis | **37.47** | 57.17 |
| | PDEBench-Comp-HighVis | **57.88** | 131.6 |
| | PDEArena-Incomp | **77.81** | 123.0 |
| Last Step | PDEBench-Comp-LowVis | **39.03** | 59.11 |
| | PDEBench-Comp-HighVis | **71.17** | 163.5 |
| | PDEArena-Incomp | **56.26** | 76.35 |
| All average | PDEBench-Comp-LowVis | **27.08** | 40.75 |
| | PDEBench-Comp-HighVis | **30.06** | 65.19 |

Table 16: **(Comparison) Summary of Resource and Timing Metrics** for different methods.

| Resource and Timing Metrics | Ours | DPOT | MPP |
|---|---|---|---|
| Training cost [GPU hrs] | 58 | 70 | 64 |
| Rollout time per step [ms] | 8.7 | 12.0 | 25.7 |
| Model Param Size | 88M | 122M | 116M |

