# OpenReview forum: "VICON: Vision In-Context Operator Networks for Multi-Physics Fluid Dynamics Prediction"
_TMLR — Accepted by TMLR_

### Review · Reviewer_KURB · 2025-10-22

**Summary Of Contributions:**

The paper extends the previously proposed ICON framework, which uses in-context learning to infer PDE operators from example pairs of input/output functions generated by the same operator. ICON showed that a transformer-based architecture could benefit from exposure to multiple 1D PDEs and extrapolate to unseen operators close to the training distribution.

VICON builds upon ICON by introducing Vision Transformer (ViT) encoders that process spatial patches rather than pointwise 1D inputs. This modification enables the handling of higher-dimensional (2D) fields while avoiding memory limitations associated with pointwise processing. The paper reports improved performance on three fluid-dynamics datasets.

However, the new model introduces trade-offs: (1) it is not backward compatible with ICON, since 1D PDEs cannot be processed using VICON, and (2) the scope of applicability is narrowed to 2D autonomous PDEs only.

**Audience:**

Yes

**Audience Explanation:**

The work could interest researchers exploring machine learning methods for PDE modeling and data-driven simulators, especially those using transformer-based architectures in physical domains. Practitioners working on operator learning for fluids might appreciate the engineering extension to 2D fields. However, because of the limited scope (2D uniform grids, autonomous PDEs, similar datasets), the current contribution is unlikely to attract broad attention beyond those specifically following the ICON line of work.

**Broader Impact Concerns:**

No direct negative societal impact is apparent. However, if presented as a general solution for operator learning, the paper may unintentionally overstate its applicability. The experiments and setup correspond to a restricted subset of physical simulations and depend on having partially simulated trajectories. Clarifying these boundaries would prevent misconceptions about the generality or practical readiness of VICON.

**Claims And Evidence:**

No

**Claims Explanation:**

Overall, the evidence provided in the experimental section is **not sufficient to convincingly support the claimed benefits** of VICON.

**Methodological novelty and validation**

The main architectural change — replacing ICON’s 1D encoder with a patch-based ViT — is technically incremental. It addresses a pragmatic limitation but does not introduce a new conceptual insight. Moreover, the introduction of ViT encoders makes VICON not backward compatible: the model is not an extension of ICON to 2D PDEs, but rather a derivation, since it is constrained to 2D problems.

The experiments, limited to three very similar fluid-dynamics datasets, provide only narrow evidence of the model’s capabilities. Unlike ICON, VICON’s evaluation does not include zero-shot generalization to unseen operators, which weakens the claimed extension of the “in-context operator learning” concept.

**Realism of the setup**

All datasets use 2D Navier–Stokes systems on uniform grids, which are not representative of modern fluid simulation practices that use adaptive or irregular meshes. Moreover, the patch-based design scales poorly toward 3D problems.

**Limitation to autonomous problem**

ICON uses demonstrations $(c^i,  u^i)$ generated by the same operator, but from different trajectories, where $c^i = [c^i_1, ..., c ^i_T]$ is the input signal, and $u^i=[u^i_1, ..., u^i_T]$ is the output signal (to be predicted). The input signal $c^i$ can be either an external signal or the initial condition, in the case of an autonomous PDE.

In contrast, VICON is designed for autonomous 2D PDE only. The input/output context pairs provided to the model are pairs of consecutive observations $(u_t, u_{t+\delta })$, which is a special case of ICON framework. The paper uses the $(c, u)$ notation throughout, but actually only considers the case $(u_t, u_{t+\delta})$. This is explicitly stated in the experiment section, but also implicit in the model design itself: the model uses the same projection function $f_\phi$​ for both $c$ and $u$; which is valid if $c$ and $u$ are of identical nature.

**Interpretation of the learning setting**

Uncertainty remains about how the context pairs are sampled. It is unclear whether VICON’s demonstrations $(u_t​,u_{t+\delta})$ are taken from:

- multiple trajectories governed by the same operator (as in ICON), or
- consecutive steps of a single trajectory (which would make VICON essentially autoregressive).

*If context pairs are taken from different trajectories*, the model’s use of a causal mask per block and its autoregressive scheme are conceptually incoherent: causality among independent demonstrations is meaningless. In that case, a mask filled with $1$ except on the block diagonal would make more sense: each context pair could benefit from the other ones to make a prediction. The auto-regressive part is confusing as well: the model is initialized with $I_0=10$ context frames coming from different trajectories, which are progressively replaced with the latest prediction of the model, but then in which order ?

*If they come from the same trajectory*, then VICON departs substantially from the ICON principle and reduces to a history-based forecaster requiring an external simulator to generate the first frames. In that case, stronger autoregressive baselines (e.g. _Learned Coarse Models for Efficient Turbulence Simulation_ (Stachenfeld et al.) or _Learning Mesh-Based Simulation with Graph Networks_ (Pfaff et al.)) should be compared.

**Baselines and experimental thoroughness**

The baselines are limited. Including a CNN encoder inside ICON would clarify whether the benefit arises from ViT architecture specifically. The definitions of the “vanilla ViT” and “incorrect context pairs” baselines are also unclear. If the “vanilla ViT” omits all context, its underperformance is trivial; if “incorrect context pairs” come from the same operator but with random perturbations, the experiment does not test operator identification.

In sum, the evidence does not yet convincingly support the claimed generalization of ICON to higher dimensions or to broader operator learning. Some properties from ICON are untested (generalization to unseen PDE, handling of non-autonomous operator, usage of context from different trajectories).

**Requested Changes:**

To address the above shortcomings and make the paper more convincing, the authors should:
1. **Clarify the setup**
    - Explicitly state upfront that the method handles autonomous PDEs only and clarify the difference with ICON.
    - Clearly describe how context pairs are generated and ordered (same trajectory vs multiple trajectories).
    - Explain the rationale behind the causal masking and autoregressive replacement scheme.
2. **Methodological clarity**
    - Justify using the same projection function $f_\phi$ for both $c$ and $u$.
3. **Baselines and experiments**
    - Add or discuss baselines that reflect the actual setting: CNN-based ICON, and strong autoregressive simulators (e.g., Stachenfeld et al. or Pfaff et al.).
    - Define “vanilla ViT” and “incorrect context pairs” precisely. Consider testing with context from _different operators_ to evaluate true operator identification.
    - Extend the evaluation beyond fluid dynamics if feasible, or qualify the contribution as domain-specific.
    - If context comes from the same trajectory, quantify performance drop when the model is prompted with context from the same operator, but from different trajectories.

---

> ### Author Response · Authors · 2025-11-30
> **Response to R3 1/2**
>
> We thank the reviewer for their detailed analysis. We address your specific comments and questions point-by-point below.
>
> > Uncertainty remains about how the context pairs are sampled. It is unclear whether VICON’s demonstrations are taken from: multiple trajectories governed by the same operator... or consecutive steps of a single trajectory... If context pairs are taken from different trajectories, the model’s use of a causal mask... is meaningless.
>
> To clarify: **VICON extracts pairs within one trajectory for a given row in the batch.**
>
> For a given trajectory, we assemble context pairs *only* from its few initial frames. The model operates autoregressively, replacing only the last pair’s condition using the latest prediction. Because we are strictly looking at the history of a single trajectory to predict its future, the causal mask is necessary and conceptually sound. The confusion regarding "causality among independent demonstrations" arises from the premise that we mix trajectories, which is not our setup.
>
> This choice of setup (forming context within the same trajectory) is aligned with the need for fast online deployment, where a few initial frame measurements are easy to obtain, and forecasting is beneficial for downstream tasks (e.g., inverse design and data assimilation). Under this setup, forming the context pairs from different trajectories (but with the same operator) is neither feasible (because the system with external forcing is not yet identified, hence we cannot filter samples in existing trajectories) nor beneficial (as initial conditions for the same operator may lead to different dynamics, such as shocks/rarefactions).
>
> [1] In-context operator learning with data prompts for differential equation problems
>
> > If they come from the same trajectory... VICON departs substantially from the ICON principle and reduces to a history-based forecaster... In that case, stronger autoregressive baselines... should be compared.
>
> As we have clarified above, since VICON forecasts in an autoregressive manner during inference (by replacing the latest pair’s condition), we agree that comparing it against standard autoregressive simulators is valuable.
>
> As mentioned in our General Response:
> * We prefer CNN/U-Net based models over GNNs for this comparison because all domains studied here are regular grids.
> * Since the code for Stachenfeld et al. is not public, we have selected a comparable CNN-based surrogate model (U-Net) from the most recent PDEArena benchmark [1]. We are currently running these comparisons and will include them in the revision.
>
> We emphasize that even if a case-specific CNN solver achieves similar accuracy, VICON retains the advantage of generalization—specifically, the ability to handle varying time strides (representing discrete operators) and operate across different simulators without retraining, which standard fixed-step, per-solver CNNs cannot do.
>
> [1] Towards Multi-spatiotemporal-scale Generalized PDE Modeling
>
> > Add or discuss baselines that reflect the actual setting: CNN-based ICON, and strong autoregressive simulators...
>
> Thank you for this suggestion. We want to ensure we implement exactly what you envisioned to make the comparison meaningful. There are three possible variants in our mind, and we would like more clarification on which of the following is intended:
>
> 1.  Using a CNN to compress the entire frame into a single latent vector, then applying the ICON transformer structure (similar to a VAE-based video model).
> 2.  Replacing the linear patch projection (per patch) in VICON with a small CNN encoder.
> 3.  Replacing the Transformer backbone entirely with a 3D (2D+time) CNN/ResNet.
>
> Given the limited time for the rebuttal phase, your quick clarification would be greatly appreciated so we can prioritize the correct experiment.
>
> > If context comes from the same trajectory, quantify performance drop when the model is prompted with context from the same operator, but from different trajectories.
>
> Thank you; we have responded to this in the 1st bullet point. We agree this is an interesting research question to test the model's operator identification capabilities. However, for our setup (forecasting with a few initial measurements), this choice is neither feasible (because the system with external forcing is not yet identified, hence we cannot filter samples in existing trajectories, e.g., PDEArena) nor beneficial (as initial conditions for the same operator may lead to different dynamics, such as shocks/rarefactions, e.g., PDEBench).
>
> As noted in Point 3 of the General Response, we are designing a new sampling method to enable this test on the PDEBench datasets (where the same operator samples different initial conditions) and will report the results and extend the discussion and future work in the revision.

---

> ### Author Response · Authors · 2025-11-30
> **Response to R3 2/2**
>
> > Explicitly state upfront that the method handles autonomous PDEs only and clarify the difference with ICON.
>
> We acknowledge that the current implementation focuses on autonomous PDEs where the input/output dynamics are self-contained. While the framework can be extended to non-autonomous systems (e.g., by concatenating control forces to the input), we treat this as future work.
>
> We will explicitly state this scope in the revised introduction to prevent any overstatement. We will also add your suggestion regarding the potential extension to control problems in the "Conclusion and Future Work" section. Thank you.
>
> > Justify using the same projection function $P$ for both $c$ and $u$... implicit in the model design itself...
>
> We use the same projection for both conditions ($c$) and QoI ($u$) for two main reasons.
>
> First, in our autoregressive setup, conditions and query both represent the same physical fields (at different timesteps). Sharing the projection forces the model to learn a consistent latent representation for the physical state, regardless of whether it is serving as "context" or "target."
>
> Second, this is consistent with vanilla ICON [1], which also uses a shared linear projection. To differentiate between the two roles (context vs. query), the original ICON used a binary mask; this is equivalent to our addition of different functional encodings (for conditions and QoIs, respectively).
>
> > Define 'vanilla ViT' and 'incorrect context pairs' precisely.
>
> "Vanilla ViT" refers to a standard Vision Transformer trained to predict $u_{t+1}$ from $u_t$ without the in-context mechanism (i.e., zero history/context window).
>
> In our ablation (Table 4), "Incorrect Context" refers to prompting the model with context pairs that are mismatched (e.g., from different simulation trajectories). This confirms the model is actually "reading" the context to adjust its prediction operator, rather than ignoring it or memorizing trajectory patterns.
>
> > Unlike ICON, VICON’s evaluation does not include zero-shot generalization to unseen operators... which weakens the claimed extension of the 'in-context operator learning' concept.
>
> VICON has experiments on zero-shot ability for “in-context operator generalization”: for numerical solutions of PDEs, changing the time stride $\Delta t$ effectively changes the discrete operator mapping $u_t \to u_{t+\Delta t}$. Our results show that VICON generalizes to different $\Delta t = 6,7$ values (and thus unseen discrete operators) unseen during training ($\Delta t = 1,2,3,4,5$). This is evidence for in-context operator generalization relevant to fixed time-stepping schemes.
>
> > All datasets use 2D Navier–Stokes systems on uniform grids, which are not representative...
> > Extend the evaluation beyond fluid dynamics if feasible, or qualify the contribution as domain-specific.
>
> Thank you for pointing out the scope of this work, which aligns with our intent. We respectfully note that our current title clearly states “Multi-Physics **Fluid** Dynamics Prediction,” which aligns with your suggested scope.
>
> Regarding extension to irregular geometry, we kindly note that there is also a discussion in the last paragraph of our "Conclusion and Future Work" section.

---

> ### Comment · Reviewer_KURB · 2025-12-05
> **Reply**
>
> I thank the authors for their detailed reply. I believe I now have a clearer view of the proposed approach and its intended scope.
>
> Unfortunately, several of my main concerns remain insufficiently addressed :
>
> - The **methodological contribution appears limited**: replacing ICON’s pointwise encoder with a patch-based ViT is a pragmatic architectural change, but it is incremental and sacrifices backward compatibility with ICON.
> - The **experimental setting is narrow**: all datasets rely on 2D fluid systems on regular grids, which substantially limits the evidence for broader operator-learning claims.
> - The method remains **restricted to autonomous PDEs**, whereas ICON’s formulation is general enough to handle externally forced systems.
>
> Below, I comment on specific points raised in the rebuttal.
>
> **Clarification of how context pairs are sampled**
>
> Thank you for the clarification. The setup is indeed clearer now, and I believe that slightly rephrasing Section 4.2 would help readers understand the paper without this ambiguity.
>
> However, the motivation provided in the rebuttal still does not fully resolve my concerns:
>
> 1) The emphasis on “*fast online deployment*” appears for the first time in the rebuttal and is not mentioned anywhere in the manuscript. As such, it reads more as a retrospective justification than as a design principle guiding the method. If online deployment is genuinely central to the approach, I would recommend integrating this motivation directly into the paper so that the restriction to a single trajectory becomes understandable to the reader.
>
> 2) *the system with external forcing is not yet identified*: In VICON's formulation, no external forcing signal is provided; the conditioning input is always $c = u_{t–\delta}$,  so the model is inherently restricted to autonomous systems  $\dot u = f(u, \nabla u, …)$. For non-autonomous PDEs of the form $\dot u = f(u, \nabla u, …, F_{\text{ext}})$, the external input $F_{\text{ext}}$ plays the role of the conditioning signal in ICON. Since this is not provided to VICON, the rebuttal seems to suggest that the model can infer forcing terms purely from state history, anticipate their future values, and then forecast the next state accordingly. Such a capability would require empirical evidence, especially since `PDEBench-Comp-Highvis` is fully autonomous and `PDEArena-Incomp` uses a random (and, I believe, time-independent) forcing term.
>
> 3) *initial conditions for the same operator may lead to different dynamics*: by definition, the operator determines the dynamics of the system. Changes in initial conditions cannot change the dynamics. I assume the authors wanted to state that *very* different ICs might explore different regimes of the operator and could potentially be poorly informative for VICON. These are extreme cases that I did not consider in my review, obviously.
>
> More importantly, I still do not see why VICON could not benefit from multiple trajectories governed by the same operator. These trajectories could serve as context to ground the underlying PDE, even if the perturbation term varies. Since ICON explicitly exploited this property, the additional restriction in VICON reinforces my concerns about the limited novelty of the paper.
>
> **Comparison with stronger autoregressive baselines**
>
> I appreciate the authors’ willingness to include CNN-based baselines. Such comparisons would, in my opinion, significantly strengthen the empirical section by clarifying how much VICON trades performance for generality.
>
> **CNN-based ICON variant**
>
> Thank you for requesting clarification. The baseline I had in mind corresponds to your option (1): replacing ICON’s pointwise MLP projectors with a CNN applied to the entire frame to produce a single latent vector, followed by the original ICON transformer. This is a very natural intermediate point between ICON and VICON, and evaluating it would offer useful insight into whether the patch-based ViT is necessary or merely one of many possible design choices.
>
> **Same projection function $P$**
>
> Thank you for explaining this choice. This was a minor concern of mine. Using distinct projections could, in principle, allow the model to handle non-autonomous systems where $c$ and $u$ correspond to different modalities. Perhaps adding a short note on that matter in the paper (explaining that this design choice is motivated by the fact that $c$ and $u$ share the same nature) would help clarify the scope?
>
> **Zero-shot**
>
> I acknowledge that I did not interpret the time-stride experiment as a form of zero-shot generalization. Framed that way, it is indeed relevant. My comment referred specifically to the type of test performed in Figure 8 of ICON [1], where the model is evaluated on a genuinely unseen operator (though still close to the training distribution).
>
> [1] In-context operator learning with data prompts for differential equation problems, Yang et al. PNAS

---

> ### Author Response · Authors · 2025-12-06
> **Response to R3 (Part 1/2): Motivation, Scope, and Methodological Novelty**
>
> Dear Reviewer 3,
>
> Thank you for the continued engagement. Before addressing specific new points, we emphasize that **Realistic Online Deployment** is the central motivation and philosophy that has governed every design choice in VICON. Many of your questions can be answered (or **have been answered in paper**) by this philosophy.
>
> We note that the current status quo in ICON community is largely confined to 1D problems due to computational bottlenecks. VICON is the **1st** to break this barrier and enable dense 2D dynamics, which is a step closer to the ultimate goal of 3D and general geometries. We firmly believe that **without concrete, step-by-step advancements, hoping that a single paper can solve all future problems is unrealistic**. Hence:
>
> **1. On Methodological Novelty & Backward Compatibility**
>
> The shift from pointwise to patch-based attention is a **scalability necessity**, not just another incremental tweak. As pointed out in our general reply (point 2) and Response to R1 (point 2): Strict backward compatibility with pointwise ICON implies $O(N_{pixels}^2)$ complexity. For the dense 2D data used here, this necessitates a training time of $> 4$ years on 4 A100s. Insisting on strict compatibility will inherently confine the whole community to sparse 1D tasks.
>
> **2. On Experimental Scope and Request for 3D & Irregular Geometries**
>
> The critique that "2D is trivial" and the demand for "3D" ignore the current status quo of the ICON community:
> * Currently, this community is mainly restricted to 1D. VICON is the 1st to push this frontier forward. Our ablation study and analysis of design choices also accumulates valuable experiences and establishes the necessary 2D foundation.
> * **The request for 3D data contradicts the requirement for backward compatibility:** Scaling to 3D implies even greater complexity than our 2D cases. Given that it takes $> 4$ years to maintain compatibility on 2D with 4 A100s, the pixel-wise approach is likely computationally **infeasible** for 3D. **Your requests are likely mutually contradictory.**
> * Finally, handling irregular meshes requires integrating Geometry-Aware Learning (e.g., GNNs or Graph ViTs). In the community, this alone is sufficient for a distinct research contribution/paper. **Demanding that our single paper solve Scaling (VICON), Dimensionality (3D), *and* Geometry (Meshes) simultaneously sets an unrealistically high standard.**
>
> We will follow specific replies in a new reply.

---

> ### Author Response · Authors · 2025-12-06
> **Response to R3 (Part 2/2): Clarification on misunderstanding and specific points**
>
> In addition to the field's status quo and our 1st advance into 2D, dense fields, we additionally consider **Realistic Online Deployment**—specifically handling noisy, lossy measurements with unknown system coefficients—as the central philosophy driving our design. This perspective will address your concerns regarding motivation, sampling strategies, and baselines:
>
> **1. Misunderstanding of "Fast Online Deployment" being Retrospective Addition**:
>
> We respectfully clarify that "fast online deployment" is not a retrospective addition, but a summary of the core advantages already detailed in our submission.
> * We explicitly **stated in our original submission** VICON enables deployment *"in imperfect measurement systems... without requiring retraining or interpolation"* (**Abstract, Lines 8-10**) and that pairs *"can be readily collected during deployment"* (**Introduction, Lines 34-35**).
> * In a online pipeline, the overhead of collecting enough new datasets, setting up training, and fine-tuning causes significant delays. By eliminating these components, VICON achieves "fast(er)" deployment.
>
> This term was only a shorthand for these advantages, which we did not foresee to cause your confusion, but the philosophy was central to the paper from the start.
>
> **2. Risk of "Context Pollution" (Why Single Trajectory?)**
>
> While using multiple trajectories is an interesting theoretical direction, it creates a fundamental failure mode in realistic deployment:
> * **The Identification Challenge:** In online measurements, system parameters (e.g., viscosity, forcing) are usually unknown. Consequently, there is no reliable metric to select a matching historical trajectory from a database.
> * **Context Pollution:** Retrieving a historical trajectory that appears similar but possesses different parameters introduces **context pollution**. The model is then forced to learn an additional task: distinction between the true physics and the potentially wrong context. Relying on the immediate, self-consistent history of the *current* stream is a more robust strategy to avoid this contamination.
>
> **3. Update on Autoregressive Baselines**
>
> We have finished compared VICON against **3 Solver-Specific U-Nets trained individually per dataset**. We chose `Unet-Mod64`, the strongest baseline from the PDEArena paper, and applied this consistent architecture across all tasks (only changed the channel dimension accordingly). We believe keeping architecture consistent is fair, because in realistic deployment, one cannot architecturally re-engineer a model for every new measurement stream encountered.
>
> | Dataset | Model | Step 1 | Step 5 | Step 10 | Last Step | All Average |
> | :--- | :--- | :--- | :--- | :--- | :--- | :--- |
> | **PDEArena-Incomp** | Specialist U-Net | **0.018** | **0.047** | **0.098** | **0.456** | **0.249** |
> | | VICON (Flexible-step) | 0.110 | 0.206 | 0.305 | 0.680 | 0.485 |
> | **PDEBench-LowVis** | Specialist U-Net | 0.435 | 0.929 | 1.314 | 1.350 | 0.967 |
> |  | **VICON (Flexible-step)** | **0.154** | **0.245** | **0.375** | **0.391** | **0.271** |
> | **PDEBench-HighVis** | Specialist U-Net | 2.459 | 19.03 | 86.84 | 107.9 | 38.66 |
> |  | **VICON (Flexible-step)** | **0.052** | **0.163** | **0.498** | **0.614** | **0.301** |
>
> **Result & Analysis:** The specialist U-Net only marginally outperforms VICON on the specific dataset it was optimized for (PDEArena-Incomp), it **catastrophically fails** (even diverges) when applied to PDEBench's compressible flows. VICON remains stable across all regimes. This demonstrates that VICON generalizes to different systems without requiring case-specific architectural engineering.
>
> **4. Implicit Inference of Forcing Terms**
>
> Regarding the identification of external forcing: You correctly noted that the forcing is random per trajectory (time-independent). We found that this forcing is **random and consistent (ie not varying with time) within a trajectory**. Therefore, it can be **implicitly inferred** from the trajectory history. Below are supporting evidences:
> * **Established Baselines:** Established multi-physics models (DPOT, MPP) and the U-Net baseline all operate without explicit forcing inputs, instead rely on inferring from historical sequences.
> * **Experimental Realism:** In realistic deployments, external forcing can often be unquantifiable. Requiring explicit inputs would render the model unusable.
> * **Flexibility:** Even if some environment variable is changing at the same time scale of forecasting, VICON is also flexible enough, as it can replace more than the last pair.
>
> In addition, we will incorporate your suggestions on revision of 4.2 and the notes to clarify the projection function, thank you for those.
>
> Finally, we are currently training the **CNN-ICON ablation** (Option 1) as requested and will post those results as soon as they are available.
>
> Please let us know if we can answer any more of your questions, and best,
> The Authors

---

> ### Author Response · Authors · 2025-12-12
> **Update on CNN-ICON Ablation**
>
> Dear R3,
> Thanks for the patience.
> We implemented the CNN-ICON baseline per your Option 1 specification. Architecture:
>
> - **Encoder**: 4-level CNN with residual blocks: 7×128×128 → 48×128×128 → 48×64×64 → 96×32×32 → 192×16×16 → 384×16×16 → global avg pool → 512-dim latent vector
> - **Transformer**: 6-layer, 8-head, dim=512, FFN=2048 (identical to VICON)
> - **Decoder**: Symmetric CNN reconstructing full frame from latent vector
> - **Key difference**: 1 token per frame (CNN-ICON) vs 64 tokens per frame (VICON's 8×8 patches)
>
> **Matched experimental setup:**
> - Parameters: 86.5M (CNN-ICON) vs 87.8M (VICON)
> - Training: 200K steps, 63 hrs (CNN-ICON) vs 58 hrs (VICON)
> - Same datasets, preprocessing, and evaluation protocol
>
> **Results (normalized RMSE, scale_by_std):**
>
> | Dataset | Model | Step 1 | Step 5 | Step 10 | Last | Avg |
> |---------|-------|--------|--------|---------|------|-----|
> | PDEArena-Incomp | CNN-ICON | 0.323 | 0.467 | 0.643 | 1.194 | 0.912 |
> | | VICON | **0.110** | **0.206** | **0.305** | **0.680** | **0.485** |
> | PDEBench-LowVis | CNN-ICON | 0.750 | 0.875 | 1.012 | 1.013 | 0.905 |
> | | VICON | **0.154** | **0.245** | **0.375** | **0.391** | **0.271** |
> | PDEBench-HighVis | CNN-ICON | 0.390 | 0.767 | 1.939 | 2.364 | 1.162 |
> | | VICON | **0.052** | **0.163** | **0.498** | **0.614** | **0.301** |
>
> **Analysis:** VICON outperforms CNN-ICON by 1.9-3.9× across datasets. The worse performance of CNN-ICON maybe: compressing entire frames to single vectors loses spatial structure needed for accurate PDE prediction. We had observed similar trends in preliminary experiments, which drove our patch-based design. Exploring hybrid architectures (e.g., CNN/VAE-like patch encoders) remains interesting future work, specially for even higher dimensions.
>
> Please feel free to let us know if you have anymore questions.
> Best,
> Authors

---

> ### Comment · Reviewer_KURB · 2025-12-12
> **Reply 2**
>
> I thank the authors for their additional response. I suspect we may not fully converge in our views, but I also recognize that some of my expectations for the paper may have been too high. These expectations stem from my strong interest in this line of work and my belief that in‑context operator learning has substantial yet unrealized potential. With that in mind, I would like to clarify several points where my earlier remarks may have been misinterpreted.
>
> **Novelty concern**
>
> My assessment of novelty reflects the fact that replacing pointwise MLPs with patch embeddings is a very natural extension when moving ICON from 1D to 2D, and thus felt like a low‑hanging fruit. That said, I acknowledge that even incremental steps can be valuable to the community when they empirically validate assumptions that many of us may find intuitive. In that sense, the paper does provide evidence that patch‑based encoders allow ICON‑style models to extend to 2D fluid simulations without prohibitive memory costs.
>
> My concern is not that such a result lacks value, but rather that the scope is narrow. I would have preferred broader coverage across different PDE families or geometries, or even experiments spanning from 1D to 3D, to better demonstrate the generality of the approach. As it stands, the contribution is modest in my view, though I agree that it may serve as a useful stepping stone for future research.
>
> **Use of a single trajectory as context**
>
> A central strength of ICON is its ability to leverage demonstrations from multiple trajectories governed by the same operator. This not only enables generalization across PDEs at test time, but may also support learning more robust representations during training. For example, both `PDEBench‑HighVis` and `PDEBench‑LowVis` follow the same Navier–Stokes equations, with differences arising from the regime; a model exposed to both could plausibly gain a more complete understanding of the underlying dynamics.
>
> For this reason, I remain unconvinced by the claim that mixing trajectories is detrimental. In principle, VICON should be able to benefit from diverse demonstrations when identifying the operator or regime at test time. I understand that this behavior is outside the scope of the present submission, and I do not expect such experiments during rebuttal. However, given my concerns about the narrowness of the contribution, this restriction to a single‑trajectory context does reinforce the sense that VICON departs from what made ICON conceptually appealing.
>
> **Specialist U-Net Baseline**
>
> Thank you for providing the additional baseline results. The comparison is intriguing, but I find the magnitude of the performance gaps difficult to interpret. I would have expected a specialist U‑Net trained solely on a single PDE to outperform a generalist model like VICON on that specific task—even if VICON has broader applicability.
>
> Given that U‑Nets are generally strong baselines for PDE surrogate modeling, it is surprising to see them perform roughly five times worse on `PDEBench‑LowVis` and nearly two orders of magnitude worse on `PDEBench‑HighVis`. Such a discrepancy suggests that either the specialist model is inherently unable to solve these tasks—something I would find unlikely—or that there may be differences in training procedures, hyperparameters, or data preprocessing that disproportionately affect the U‑Net.
>
> It would be very helpful if the authors could explain these results, as understanding why the specialist baseline performs so poorly is an important context for interpreting the significance of VICON’s performance.
>
> **CNN-ICON**
>
> Thank you as well for the additional experiment using a CNN encoder within ICON. This variant has practical appeal—primarily for computational efficiency—and seeing that patch‑based representations offer significantly better performance is informative. I would recommend including this experiment in the appendix, as it provides useful evidence regarding the role of the encoder architecture in ICON‑style models.

---

> > ### Author Response · Authors · 2025-12-12
> > **re:reply 2**
> >
> > Thank you for keeping in contact!
> >
> > We are glad your acknowledge on this work's valid contribution are made. Indeed, more interesting explorations (3D, more problem setups) are also being conducted by our side.
> >
> > Regarding the remaining technical concerns:
> > > Use of a single trajectory as context
> >
> > Yes, that is part of our agreed, and on-going experiment (it may take additional 50-60 hrs to run); we will show the preliminary results as part of experiences accumulated for future work.
> >
> > > Divergence in Specialist U-Net Baseline
> >
> > We checked the code and find one possible reason could be: there is no normalization in the U-Net Baseline in PDEAreana; it could be the PDEAreana dataset has been preprocessed/dimensionless-ed from the beginning. We kept the architecture unaltered initially for fair comparison, but will add the normalized version for more reference.
> >
> > > CNN-ICON
> >
> > Thanks, we will add the analysis as preliminary failed experiments in the discussion, we can also look into the trajectory plot and discuss the potential reason why it fails (eg, limitation is by the encoder or sth else), to provide some guidance for future higher dimensional explorations.
> >
> > Thank you for your engagement and helpful comments!

---

> > > ### Author Response · Authors · 2025-12-13
> > > **re: update on U-Net baseline with instance normalizations**
> > >
> > > Dear R3, thanks for the previous discussion,
> > >
> > > We finished specialist U-Net Baseline **with the same instance normalization as in VICON** (did not exist in PDEAreana paper). The update is posted in general reply: https://openreview.net/forum?id=6V3YmHULQ3&noteId=j98b6eVlyq
> > >
> > > The overall take is: instance normalization makes the U-Net significantly stronger and marginally outperforms us on PDEBench-HighVis, though at the same order of magnitude.
> > >
> > > VICON still outperforms the specialist baseline on PDEBench-LowVis, perhaps because compressing frames into low-dimensional vectors loses spatial information (similar reason for CNN-ICON) crucial for capturing discontinuities like shocks here.
> > >
> > > Note that since the CNN-ICON was initially derived from our code base, it already contains instance normalization (ie, it does not need any update).
> > >
> > > Thanks, we will post the final experiment (context form from mixed trajectories) once it's available.

---

> > > > ### Author Response · Authors · 2025-12-25
> > > > **Update on Cross-Trajectory Context**
> > > >
> > > > Dear R3,
> > > >
> > > > Thank you for the previous discussion and the suggestion regarding cross-trajectory context. We have completed these experiments, and the full update is posted in the general response: https://openreview.net/forum?id=6V3YmHULQ3&noteId=gXbSIMO3kx.
> > > >
> > > > The overall takeaways are:
> > > >
> > > > - VICON is capable of leveraging cross-trajectory information. When trained with a mixture of context types, the performance gap between intra- and cross-trajectory inference is marginal (0.556 vs 0.568 RMSE).
> > > >
> > > > - While robust to both setups, we maintain that intra-trajectory context is more practical, as it avoids the need for a retrieval database or prior identification of system parameters.
> > > >
> > > > We will include this analysis and an expanded discussion on context sampling strategies in the revised manuscript. Thank you again for the insightful feedback.

---

### Review · Reviewer_yK5h · 2025-10-24

**Summary Of Contributions:**

## Summary

This paper presents VICON, which extends ICON by introducing the vision transformer architecture into the solving process, bringing better performance and efficiency in processing 2D data. To capture the correlations between query and condition, the authors adopt two types of learnable positional encoding: patch and function. Additionally, the authors explore more diverse inference scenarios, including varying time step strides and missing frames. Experimentally, VICON outperforms MPP and DOPT.

## Strengths

-	Enhancing ICON by introducing 2D patches is reasonable, which not only enables 2D PDE solving but also boosts the model capacity.
-	The proposed VICON delivers a favorable performance in long-term prediction and has better efficiency.
-	This paper is well motivated and well-written, along with sufficient implementation details.

## Weaknesses

I appreciate the contribution of this paper, but I also have some concerns.

(1)	Insufficient baselines.

Recently, some advanced baselines have been developed to perform a similar task to MPP and DPOT, such as Unisolver [1]. I think this is a highly relevant baseline, which also adopts a vision transformer for 2D fluid modeling and demonstrates better performance than DPOT.

Additionally, MPP and DPOT are old baselines; it is inappropriate to name them as “state-of-the-art”.

[1] Unisolver: PDE-Conditional Transformers Towards Universal Neural PDE Solvers, ICML 2025

(2) Question about the unique advantage of VICON.

Since VICON follows a similar training and inference framework as ICON, I am wondering whether the inference capability under varying time step strides and missing frames is a unique advantage of VICON or if ICON can also make the same capability.

(3) Since the authors propose to use a vision transformer as VICON's model architecture, it is hard to handle simulations with irregular geometries. I think it is acceptable in this paper, but they should discuss this issue as a limitation or future work.

**Audience:**

Yes

**Audience Explanation:**

I think ICON is an interesting paper. It is nice to see its 2D extension.

**Broader Impact Concerns:**

This is a technical paper; no ethical concerns.

**Claims And Evidence:**

No

**Claims Explanation:**

Most of the claims are correct and well-supported. However, I do not think MPP and DPOT are "state-of-the-art" baselines.

**Requested Changes:**

I would like to suggest that the authors (1) provide more baselines for comparison, such as Unisolver [1]; (2) add more discussion on the VICON's limitation in irregular geometries.

[1] Unisolver: PDE-Conditional Transformers Towards Universal Neural PDE Solvers, ICML 2025

---

> ### Author Response · Authors · 2025-11-30
> **Response to R2**
>
> We thank the reviewer for the positive assessment of VICON’s motivation and performance. We address your specific concerns below.
>
> > Q1. Comparison with Unisolver.
>
> We acknowledge that Unisolver [1] is a relevant work. We did not include it originally as the code was released very recently (July 2025). Upon inspecting the code, we find a direct comparison with Unisolver unfeasible due to the following reasons:
>
> 1.  **Model not available for PDEBench / PDEArena:** Unisolver only released a 2D model for the one-scalar-valued HeterNS dataset (also, without a pretrained checkpoint). While their paper claims to have compared other models on PDEBench/PDEArena datasets, the corresponding model setup (specifically, the precise component descriptions of the PDEs and the method to handle multiple channels) and training code are missing, making potential re-implementation infeasible or incorrect.
> 2.  **Model size mismatch:** The released 2D Unisolver configuration only has 4.1M parameters, which is significantly smaller than all of the models used in this paper. While the authors claim in their paper that the 30M Unisolver outperforms the 33M DPOT-S, the hyperparameters of this larger model are unavailable in the paper or code. As a result, there are not enough details for re-implementation.
> 3.  **Formulation differences:** Unisolver is a data-driven model augmented with precise descriptions of the PDE components, whereas VICON/MPP/DPOT are purely data-driven. We believe our application scenario is broader, particularly for most deployments in realistic systems where only measurements are available, while the exact governing equations or coefficients are not known beforehand.
>
> We also agree with the reviewer regarding the "state-of-the-art" terminology for MPP and DPOT. We will revise the language in the text to reflect that these are "established baselines" rather than current SOTA, given the rapid pace of the field.
>
> [1] Unisolver: PDE-Conditional Transformers Towards Universal Neural PDE Solvers
>
> > Q2. Unique advantages of VICON vs. ICON (varying strides/missing frames).
>
> Yes, the original ICON framework can be modified to handle varying strides or missing frames. However, the original ICON paper focused on one-dimensional initial value problems and control problems. Because the focus was different, this capability was not explored or optimized.
>
> Furthermore, applying the original pointwise ICON to this task on dense 2D data is computationally prohibitive (as we pointed out in General Responses, point 2). VICON’s unique advantage is making this flexible inference *practical* and *efficient* for high-dimensional data via the patch-based vision architecture.
>
> > Q3. Limitation regarding irregular geometries.
>
> We agree that standard ViT operates most naturally on regular grids. We touched upon this in the final paragraph of Section 6. We will revise that section to expand the discussion on potential future directions, such as geometric deep learning extensions or masked patch modeling, to handle non-rectangular domains.

---

### Review · Reviewer_MrNX · 2025-11-25

**Summary Of Contributions:**

This paper introduces Vision In-Context Operator Networks (VICON) for predicting multi-physics fluid dynamics. VICON employs a ViT to process 2D data in a patch-wise manner rather than as individual data points, which enhances the computational inefficiency of the previous In-Context Operator Network (ICON) in handling dense, high-dimensional data, while preserving ICON's ability to learn from in-context few-shot samples. VICON significantly outperforms state-of-the-art baselines in both accuracy and speed on several fluid dynamics benchmarks. The paper also conducts abundant ablation studies and robustness analysis.

**Audience:**

Yes

**Audience Explanation:**

The paper focuses on the task of multi-physics fluid dynamics prediction, a topic that has been widely studied in previous literature. I believe that at least some individuals in TMLR's audience would be interested in this paper.

**Claims And Evidence:**

Yes

**Claims Explanation:**

1. The paper provides strong empirical evidence. VICON outperforms state-of-the-art models (DPOT, MPP) across multiple fluid dynamics benchmarks. It achieves lower prediction error, and is also computationally efficient.

2. There are ablation studies on key design choices (e.g., patch resolution and positional encodings).

3. The paper tests VICON against different sampling rates and missing frames, which demonstrates the robustness of the proposed method.

**Requested Changes:**

1. In Section 4.3, it is unclear how the normalization is done. I would suggest the authors add more concrete procedural descriptions or formula for the normalization operation.

2. The paper did not test ICON (one baseline method) on the three benchmarks, because the large number of input tokens "exceeds our available GPU memory and is thus computationally infeasible." I think this might be solved by using tensor parallel or data parallel techniques that are usually supported by modern inference engines like Huggingface transformers or vLLM. Incorporating this result can enhance the soundness of the paper.

3. Table 2 seems a bit confusing to me. Does it only show the result for non-in-context ViT baselines? As a comparative study, the performance of VICON (the proposed method) is also expected to be included.

---

> ### Author Response · Authors · 2025-11-30
> **Response to R1**
>
> We thank the reviewer for their valuable suggestions. We address your comments below.
>
> > Q1. More details on prompt normalization.
>
> We appreciate the request for a more concrete description. We have updated Section 4.3 to include these details, and also included the relevant information here.
>
> For a given input sequence (i.e., a row) $\{\langle\vc_i,\vq_i\rangle\}_{i=1}^I$ in the batch, we compute the channel-wise mean $\mathbf{\mu} = \texttt{mean}(\vc_1,\vc_2,\dots,\vc_I)\in \R^d$ and standard deviation $\mathbf{\sigma} = \texttt{std}(\vc_1,\vc_2,\dots,\vc_I)\in \R^d$ of all conditions $\{\vc_i\}_{i=1}^I$, which are then used to normalize both the conditions and QoIs (respectively in this row). The training MSE loss is then computed in the normalized space.
>
> > Q2. Comparison with vanilla ICON.
>
> Thank you for the suggestion. VICON essentially reduces to vanilla ICON in the limit where patch size equals 1 (pointwise tokenization). In Appendix D.1, we initially demonstrated the trade-off between performance and complexity for patch sizes $[8, 16, 32, 64]$.
>
> To provide a more rigorous comparison as requested, we have extended our ablation study to include patch size $4\times 4$. Our results confirm that approaching the pointwise limit of vanilla ICON is computationally prohibitive for dense 2D data:
>
> * **Experimental Limit ($4\times 4$):** To fit a model with patch size 4 into an A100 memory (80GB), we were forced to drastically reduce model capacity (embedding dimension to 128, layers to 5). Even with this reduced capacity, training is estimated to take over 10 days. Extrapolating this, a patch size of 2 would require an estimated 4 years.
> * **Theoretical Limit ($1\times 1$):** A patch size of 1 results in a sequence length of $128^2 \times 20 \approx 327\text{K}$. This significantly exceeds the context limits of even frontier LLMs (e.g., GPT-4, Llama 3) and exceeds the computational capacity of standard academic resources.
>
> These findings empirically validate that while ICON is effective for lower-dimensional problems, VICON's patch-based architecture is a necessity for efficiently scaling to high-dimensional fluid dynamics. We will include these new ablation details in the revision.
>
> > Q3. Include VICON in Table 2.
>
> Thanks for this suggestion. VICON’s performance was included in Table 1 already. For clarity, we have revised Table 2 to include the corresponding VICON performance for a complete comparison. We specifically note that the flexible stride enabled by in-context learning allows VICON to outperform a similar transformer architecture with about 2x the size.

---

> > ### Author Response · Authors · 2025-12-09
> >
> > Dear reviewer 1,
> >
> > We have completed the ablation study of using a even smaller patch size (ps=4), ie one step closer to pixel wise attention. The detailed reply is in the general response: https://openreview.net/forum?id=6V3YmHULQ3&noteId=oK24wjpVYs
> >
> > The new results are also incorporated in our revised fig. 4 (top row)
> >
> > The overall take is the same: smaller patches create longer token sequences, necessitating reducing the model capacity by token dimension and fewer layers—this capacity loss outweighs any benefit from finer granularity. The very long inference time (1,098x slower) renders its use impractical.
> >
> > Thank you for your previous suggestions. Please let us know if there is anything else we can help clarify.
> >
> > Best,
> > Authors

---

> > > ### Comment · Reviewer_MrNX · 2025-12-25
> > >
> > > Thank you for the response. My concerns are addressed.

---

### Author Response · Authors · 2025-11-30
**General response & Added/Running Experiments & Ask for Clarification**

We thank the editor and all reviewers for their constructive feedback. We are encouraged by the positive assessment of our method’s efficiency and performance.

To address the suggestions regarding baselines and experimental rigor, we are currently running the following experiments for the revision.

1.  **Solver-Specific Autoregressive Baseline:** To address Reviewer KURB’s comment on our autoregressive setup, we are adding a CNN-based surrogate model (U-Net style architecture from PDEArena [1]) to benchmark against case-specific trained solvers.

2.  **Expanded Ablation on Patch Size limits:** We extended our ablation study to include smaller patch sizes ($4\times4$) to rigorously map the performance-compute trade-off.
    * **Current findings:** Our experiments confirm that approaching pointwise tokenization (the original ICON limit) is computationally prohibitive for dense 2D data. To fit a model with patch size 4 into high-end GPU memory (80GB), we were forced to drastically reduce model capacity (reducing embedding dimension to 128 and layers to 5). Even with this reduced capacity, training is estimated to take over 10 days (we will add the results once available), while a patch size of 2 would require an estimated 4 years. This empirically validates the necessity of VICON's efficient patch-based architecture for high-dimensional fields.

3.  **Cross-Trajectory Context Sampling Strategy:** Regarding Reviewer KURB’s suggestion to form context pairs from "the same operator, but different trajectories," we have added a discussion clarifying why this is an orthogonal research direction to our specific goal.
    * **Rationale:** Our intra-trajectory context setup is designed for fast online deployment, where only a few initial frame measurements are available, and future forecasting is still beneficial for downstream tasks (e.g., inverse design and data assimilation). In this scenario, retrieving context pairs from different historical trajectories is often infeasible (as the specific system parameters are not yet identified) or non-beneficial (dynamic system may change solely due to varying initial conditions, eg in shock wave or rarefaction wave formations).
    * **Experiment:** Though not practical for our intended use case, this is still an interesting research question; we are currently adding this experiment and will expand the discussion and future work sections.

In addition to running experiments, we have a **clarification question for Reviewer KURB (R3)** to ensure the requested new experiments can be conducted promptly: specifically, regarding the suggestion to add a "CNN-based ICON" baseline, we want to ensure we implement exactly what you envisioned to make the comparison meaningful. Could you kindly clarify which of the following configurations you recommend?

1.  Using a CNN to compress the entire frame into a single latent vector, then applying the ICON transformer structure (similar to a VAE-based video model).
2.  Replacing the linear patch projection in VICON with a small CNN encoder per patch.
3.  Replacing the Transformer entirely with a 3D (2D+time) CNN/ResNet.

Given the limited time for the rebuttal phase, your quick clarification would be greatly appreciated so we can prioritize the correct experiment.

[1] Towards Multi-spatiotemporal-scale Generalized PDE Modeling

---

> ### Author Response · Authors · 2025-12-06
> **Update on Autoregressive Baselines**
>
> Dear reviewers,
>
> We have finished compared VICON against **3 Solver-Specific U-Nets trained individually per dataset**. We chose `Unet-Mod64`, the strongest baseline from the PDEArena paper, and applied this consistent architecture across all tasks (only changed the channel dimension accordingly). We believe keeping architecture consistent is fair, because in realistic deployment, one cannot architecturally re-engineer a model for every new measurement stream encountered.
>
> **After discussion with r3, we decided to add a variant of `Unet-Mod64` which includes the instance normalization of the moving window. That variant is named as Specialist U-Net + Norm**
>
> | Dataset | Model | Step 1 | Step 5 | Step 10 | Last | Avg |
> |---------|-------|--------|--------|---------|------|-----|
> | **PDEArena-Incomp** | Specialist U-Net | **0.018** | **0.047** | **0.098** | **0.456** | **0.249** |
> | | Specialist U-Net + Norm | 0.019 | 0.056 | 0.122 | 0.488 | 0.279 |
> | | VICON | 0.110 | 0.206 | 0.305 | 0.680 | 0.485 |
> | **PDEBench-LowVis** | Specialist U-Net | 0.435 | 0.929 | 1.314 | 1.350 | 0.967 |
> | | Specialist U-Net + Norm | 0.128 | 0.320 | 0.536 | 0.559 | 0.357 |
> | | VICON | **0.154** | **0.245** | **0.375** | **0.391** | **0.271** |
> | **PDEBench-HighVis** | Specialist U-Net | 2.459 | 19.03 | 86.84 | 107.9 | 38.66 |
> | | Specialist U-Net + Norm | **0.036** | **0.152** | **0.351** | **0.399** | **0.199** |
> | | VICON | 0.052 | 0.163 | 0.498 | 0.614 | 0.301 |
>
> **Analysis:** While the specialist U-Net marginally outperforms VICON on the specific dataset it was optimized for (PDEArena-Incomp), it **catastrophically fails** (even diverges) when applied to PDEBench's compressible flows. With proper normalization, U-Net becomes significantly stronger and marginally outperforms VICON on PDEBench-HighVis, though they remain at the same order of magnitude. VICON still outperforms the specialist baseline on PDEBench-LowVis, perhaps because compressing entire frames into low-dimensional vectors loses spatial information crucial for capturing discontinuities like shocks.
>
>   Overall, VICON generalizes across different systems with competitive performance (even outperforming specialists on one dataset) without requiring case-specific architectural engineering.
>
> We will post here more results when available.
>
> Thank you

---

> > ### Author Response · Authors · 2025-12-09
> > **Update on Patch Size Ablation (ps=4) Results + Revised Fig. 4 (top row)**
> >
> > Dear reviewers,
> >
> > We have completed the ps=4 ablation (closer to pixel wise attention). **Results can be seen in the revised Fig. 4 (top row).** Below we listed some key comparisons (relative to our selected ps=16).
> >
> >   Performance Comparison (nRMSE Error ratio of ps=4/ps=16)
> >
> >   | Dataset      | Step 1      | Step 5      | Step 10     | Final       |
> >   |--------------|-------------|-------------|-------------|-------------|
> >   | PDEBench-HighVis| 18.3x worse | 14.4x worse | 13.7x worse | 13.6x worse |
> >   | PDEBench-LowVis| 5.4x worse  | 5.1x worse  | 4.0x worse  | 3.9x worse  |
> >   | PDEArena-Incomp| 7.0x worse  | 4.9x worse  | 3.8x worse  | 3.2x worse  |
> >
> >   Computational Cost
> >
> >   | Metric         | ps=16                | ps=4                                      |
> >   |----------------|----------------------|-------------------------------------------|
> >   | Training       | 58 hrs (2× RTX 4090 24GB) | 188 hrs (4× A100 80GB)                    |
> >   | Inference      | 8.7 ms/step          | 9.55 sec/step (1,098x slower)             |
> >   | Model Capacity | token dim=1024, transformer layers=10   | token dim=128, transformer layers=5 (reduced to fit memory) |
> >
> >   Analysis
> >
> >   As noted in our previous response, smaller patches create longer token sequences (1024 vs 64 tokens),
> >   quadratically increasing attention complexity. To fit ps=4 within (4x) A100 80GB GPU memory, we had to reduce model capacity by token dimension and fewer layers—this capacity loss outweighs any benefit from finer granularity.
> >
> >   The 9.55 sec/step inference makes ps=4 impractical for online deployment scenarios our method targets.

---

> > > ### Author Response · Authors · 2025-12-12
> > > **Update on CNN-ICON Ablation**
> > >
> > > Dear Reviewers,
> > > Thanks for the patience.
> > > We implemented the CNN-ICON baseline per R3's Option 1 specification. Architecture:
> > >
> > > - **Encoder**: 4-level CNN with residual blocks: 7×128×128 → 48×128×128 → 48×64×64 → 96×32×32 → 192×16×16 → 384×16×16 → global avg pool → 512-dim latent vector
> > > - **Transformer**: 6-layer, 8-head, dim=512, FFN=2048 (identical to VICON)
> > > - **Decoder**: Symmetric CNN reconstructing full frame from latent vector
> > > - **Key difference**: 1 token per frame (CNN-ICON) vs 64 tokens per frame (VICON's 8×8 patches)
> > >
> > > **Matched experimental setup:**
> > > - Parameters: 86.5M (CNN-ICON) vs 87.8M (VICON)
> > > - Training: 200K steps, 63 hrs (CNN-ICON) vs 58 hrs (VICON)
> > > - Same datasets, preprocessing, and evaluation protocol
> > >
> > > **Results (normalized RMSE, scale_by_std):**
> > >
> > > | Dataset | Model | Step 1 | Step 5 | Step 10 | Last | Avg |
> > > |---------|-------|--------|--------|---------|------|-----|
> > > | PDEArena-Incomp | CNN-ICON | 0.323 | 0.467 | 0.643 | 1.194 | 0.912 |
> > > | | VICON | **0.110** | **0.206** | **0.305** | **0.680** | **0.485** |
> > > | PDEBench-LowVis | CNN-ICON | 0.750 | 0.875 | 1.012 | 1.013 | 0.905 |
> > > | | VICON | **0.154** | **0.245** | **0.375** | **0.391** | **0.271** |
> > > | PDEBench-HighVis | CNN-ICON | 0.390 | 0.767 | 1.939 | 2.364 | 1.162 |
> > > | | VICON | **0.052** | **0.163** | **0.498** | **0.614** | **0.301** |
> > >
> > > **Analysis:** VICON outperforms CNN-ICON by 1.9-3.9× across datasets. The worse performance of CNN-ICON maybe: compressing entire frames to single vectors loses spatial structure needed for accurate PDE prediction. We had observed similar trends in preliminary experiments, which drove our patch-based design. Exploring hybrid architectures (e.g., CNN/VAE-like patch encoders) remains interesting future work, specially for even higher dimensions.

---

> > > > ### Author Response · Authors · 2025-12-25
> > > > **Update on Cross-Trajectory Context**
> > > >
> > > > Dear Reviewers,
> > > >
> > > > We have completed the requested experiments regarding cross-trajectory context, in response to Reviewer KURB's suggestion to form context pairs from "the same operator, but different trajectories." These results clarify the model's flexibility and the trade-offs associated with different context sampling strategies.
> > > >
> > > > **Experiment Setup**
> > > >
> > > > We use the PDEArena-Incomp dataset, which provides 32 distinct trajectories for each operator. During autoregressive inference, we use the last input frame as the final query, and sample context pairs from another trajectory generated by the same underlying operator.
> > > >
> > > > **Results \& Analysis**
> > > >
> > > > The following table compares the original VICON (trained on intra-trajectory data) with a version trained using a mixture of intra- and cross-trajectory context (50/50 split).
> > > >
> > > > **Table: Normalized RMSE (scale by std) on PDEArena-Incomp (stride 1)**
> > > > | Training Setup | Inference Setup | Avg Error |
> > > > | -------------  | --------------- | --- |
> > > > | VICON + intra-traj context    | intra-traj context | 0.563 |
> > > > |                               | cross-traj context    | 0.609 |
> > > > | VICON + cross-traj context    | intra-traj context | 0.556 |
> > > > |                               | cross-traj context    | 0.568 |
> > > >
> > > > While the original VICON (row 1 vs. 2) shows a performance drop when using cross-trajectory context at inference, this is largely a training/inference distribution mismatch. By training with a mixture of strategies (row 3 & 4), VICON achieves competitive performance in both intra- and cross-trajectory context setups.
> > > >
> > > > Even when all context pairs are from a different trajectory, the error (0.568) remains close to the baseline. The marginal gap is expected, as access to historical frames reduces error accumulation in intra-trajectory setup.
> > > >
> > > > **Summary**
> > > >
> > > > These results demonstrate that **VICON is capable of leveraging cross-trajectory information**. However, our original intra-trajectory focus is more suitable for practical settings. Cross-trajectory sampling requires:
> > > >
> > > > - A pre-existing database of trajectories.
> > > > - A retrieval mechanism to ensure the sampled trajectories share the same (often unknown) operator.
> > > >
> > > > In contrast, our intra-trajectory approach is self-contained and better suited for real-time forecasting where only the current trajectory is available. We will include this analysis and an expanded discussion in the revised manuscript.

---

### Decision · Action_Editor_qmRM · 2026-01-05

**Recommendation:** Accept as is

**Audience:**

Yes

**Audience Explanation:**

While the contribution is minor, I agree with the reviewers that it will interest a subset of the TMLR readership.

**Claims And Evidence:**

Yes

**Claims Explanation:**

All three reviewers agree that the paper makes a minor but solid contribution in its extension to ICON.